# Inactivation of *Sirt6* ameliorates muscular dystrophy in *mdx* mice by releasing suppression of utrophin expression

Angelina M. Georgieva [1], Xinyue Guo[1], Marek Bartkuhn[2], Stefan Günther[1], Carsten Künne [1], Christian Smolka [1], Ann Atzberger [1], Ulrich Gärtner[3], Kamel Mamchaoui[4], Eva Bober[1], Yonggang Zhou[1], Xuejun Yuan [1✉] & Thomas Braun [1✉]

The NAD$^+$-dependent SIRT1-7 family of protein deacetylases plays a vital role in various molecular pathways related to stress response, DNA repair, aging and metabolism. Increased activity of individual sirtuins often exerts beneficial effects in pathophysiological conditions whereas reduced activity is usually associated with disease conditions. Here, we demonstrate that SIRT6 deacetylates H3K56ac in myofibers to suppress expression of utrophin, a dystrophin-related protein stabilizing the sarcolemma in absence of dystrophin. Inactivation of *Sirt6* in dystrophin-deficient *mdx* mice reduced damage of myofibers, ameliorated dystrophic muscle pathology, and improved muscle function, leading to attenuated activation of muscle stem cells (MuSCs). ChIP-seq and locus-specific recruitment of SIRT6 using a CRISPR-dCas9/gRNA approach revealed that SIRT6 is critical for removal of H3K56ac at the Downstream utrophin Enhancer (DUE), which is indispensable for utrophin expression. We conclude that epigenetic manipulation of utrophin expression is a promising approach for the treatment of Duchenne Muscular Dystrophy (DMD).

[1] Department of Cardiac Development and Remodeling, Max Planck Institute for Heart and Lung Research, 61231 Bad Nauheim, Germany. [2] Biomedical Informatics and Systems Medicine, Justus Liebig University, Giessen, Germany. [3] Institute for Anatomy and Cell Biology, University of Giessen, Giessen, Germany. [4] Sorbonne Université, Inserm, Institut de Myologie, Centre de Recherche en Myologie, F-75013 Paris, France. ✉email: xuejun.yuan@mpi-bn.mpg.de; thomas.braun@mpi-bn.mpg.de

Duchenne muscular dystrophy (DMD) is an X-linked genetic disorder caused by a loss-of-function mutation in the *DMD* gene coding for dystrophin, a 427 kDa sub-membrane cytoskeletal protein[1]. Dystrophin is associated with the inner surface of the sarcolemma and part of a large macro-molecular complex of proteins, the dystrophin-associated protein complex, that stabilizes the sarcolemma and plays a key role in the regulation of multiple signaling processes. Absence of dystrophin results in increased membrane fragility and permanent damage of myofibers, prompting continuous rounds of muscle degeneration and regeneration. During disease progression skeletal muscle mass decreases rapidly in DMD patients and muscle tissue is replaced with fibrotic and fatty tissue[2].

*Mdx* mice harbor a spontaneous loss-of-function mutation in dystrophin gene, which makes them a broadly used animal model to investigate the mechanisms and pathology of DMD[3]. Compared to human DMD patients, *mdx* mice only develop a rather mild phenotype, which is in part due to increased expression of utrophin (*Utrn*)[4]. UTRN is a dystrophin-related protein that shows sequence homology and structural similarities with dystrophin. In normal adult muscles, UTRN is primarily localized at the neuromuscular and myotendinous junction but not at the sarcolemma[3,5,6]. In continuously regenerating dystrophic or injured muscle, *Utrn* is upregulated and redirected to the sarcolemma[7], although the extent of endogenous *Utrn* expression is not sufficient to completely compensate for the absence of dystrophin[8]. Importantly, transgenic overexpression of *Utrn* markedly improves muscular dystrophy in *mdx* mice, offering a potential treatment option for DMD[9,10]. Therefore, numerous efforts were made to stimulate expression of endogenous *Utrn* using small molecules[11,12]. Although this approach yielded promising results, further research is necessary, which requires more detailed insights into signaling or epigenetic mechanisms controlling *Utrn* expression. Mechanisms that remove active chromatin marks at the *Utrn* gene during development might be particularly interesting in this context, since expression of *Utrn* is strongly downregulated during the transition from embryonic to adult stages[13,14].

Mutations in the dystrophin gene lead to global transcriptional changes[15,16], which reflect cellular efforts to cope with the disease condition, but might also further aggravate progression of the disease. Such changes include epigenetic modifiers, which have attracted particular attention. In fact, previous studies uncovered that enzymes affecting histone lysine acetylation level are directly associated with DMD pathogenesis[17,18]. For example, histone acetyltransferase p300/CBP is significantly downregulated in *mdx* muscles, while expression and activity of the histone deacetylase HDAC2, which belongs to class I HDACs, are upregulated. Repression of HDAC2 activity ameliorates dystrophic changes in *mdx* muscles and promotes functional recovery[19,20]. Likewise, sirtuins, which belong to the HDAC class III deacetylase family and comprise seven members (SIRT1–SIRT7), play a major role in many human pathologies by sensing and coordinating stress responses[21]. Activation of SIRT1 leads to beneficial effects in *mdx* mice by attenuating oxidative stress and inflammation and suppressing tissue fibrosis[22–24]. So far, mainly small molecules such as resveratrol were used to directly stimulate SIRT1 activity. However, recent evidence indicates that sirtuins form a complex regulatory network, in which SIRT7 suppresses SIRT1, offering a means to indirectly control SIRT1 via SIRT7[25,26]. At present, little is known about the function of SIRT7 and its paralogue SIRT6 in skeletal muscles, which together with SIRT1 represent a subgroup of predominantly nuclear-localized sirtuins. SIRT6 plays a pivotal role in heterochromatin stabilization through deacetylation of H3K9ac, H3K18ac and H3K56ac. Germline inactivation of *Sirt6* leads to significant reduction in fiber cross-sectional diameter and increased muscle fibrosis, albeit muscle-specific ablation of *Sirt6* does not cause a strong muscle phenotype with the exception of impaired glucose homeostasis and insulin sensitivity, suggesting that adverse effects on muscle morphology in germline *Sirt6* mutants do not originate from muscle fibers[27].

In this study, we analyzed the expression of different sirtuins in muscles of *mdx* mice and detected decreased *Sirt1* but increased *Sirt6* expression both in quiescent MuSC and in muscle fibers. Pax7-Cre mediated inactivation of *Sirt6* in *mdx* mice normalized pathological features of dystrophic muscles and attenuated persistent activation of MuSC, whereas no obvious morphological and functional abnormalities were detected in skeletal muscles of *Sirt6* mutants. The absence of *Sirt6* resulted in massive hyperacetylation of H3K56, indicating that SIRT6 is the dominant H3K56ac deacetylase in skeletal muscle. Mechanistically, we found that SIRT6 suppresses *Utrn* expression in dystrophic muscles via deacetylation of H3K56ac at the downstream utrophin enhancer (DUE). We reason that upregulation of *Utrn* expression in the absence of *Sirt6* is the main cause of improved muscle function in *Sirt6^mKO/mdx* compound mutants, since inactivation of *Utrn* in *Sirt6^mKO/mdx* mice prevented the beneficial effects of the loss of *Sirt6* in *mdx* mice.

## Results

**SIRT6 is upregulated in skeletal muscles and MuSCs of mdx mice.** The continuous damage of skeletal muscles in DMD leads to myofiber degeneration and induces inflammation and fibrotic responses. In addition, muscle stem cells (MuSCs) will become activated to repair damaged fibers. To determine the impact of inflammation and chronic muscle regeneration on the transcriptional profile of MuSC, we performed RNA-seq of MuSCs isolated from wild type and *mdx* mice. We detected 2056 deregulated genes in *mdx* MuSCs compared to wild type (hereafter referred as control), with more than 125 biological processes that were significantly affected (Fig. 1a, b). GO term analysis revealed that many of deregulated genes are related to apoptosis, cell cycle, transcription, cellular stress responses, and responses to cAMP (Fig. 1b). To explore whether gene expression changes also have a functional impact, we performed TUNEL and EdU incorporation assays. Apoptosis was increased in *mdx* MuSC, probably to clear damaged and non-functional MuSCs required for an effective muscle regeneration (Supplementary Fig. 1a). In vivo EdU incorporation assays demonstrated a substantial increase of MuSCs proliferation (Supplementary Fig. 1b). The increase of cell cycle activity corresponds well to changes in cAMP signaling, which plays an important function to keep MuSCs in the quiescent state[28]. Furthermore, ~100 stress response-related genes were upregulated in freshly isolated *mdx* MuSCs, which probably evoke cellular countermeasures to master the inflammatory, adverse environment in dystrophic muscles (Supplementary Fig. 1c and Supplementary Table 1).

Since sirtuins are critical regulators of cellular stress responses, we asked whether expression or activity of sirtuins is deregulated in muscular dystrophy. Surprisingly, only *Sirt6* was significantly upregulated in *mdx* MuSCs among the nuclear sirtuins, whereas *Sirt1* was downregulated (Fig. 1c). Consistently, protein levels of *Sirt6* were markedly elevated in both MuSCs and muscle tissues of *mdx* mice (Fig. 1d). mRNA and protein levels of *Sirt6* were also markedly upregulated in DMD patient derived myoblasts carrying different dystrophin gene mutations, recapitulating the situation in *mdx* MuSCs and myofibers (Fig. 1e, f).

To investigate the role of *Sirt6* in muscle tissue homeostasis and regulation of MuSCs, we specifically inactivated *Sirt6* in the skeletal muscle lineage using mice expressing constitutively active Cre recombinase under control of the Pax7 promoter (*Pax7-*

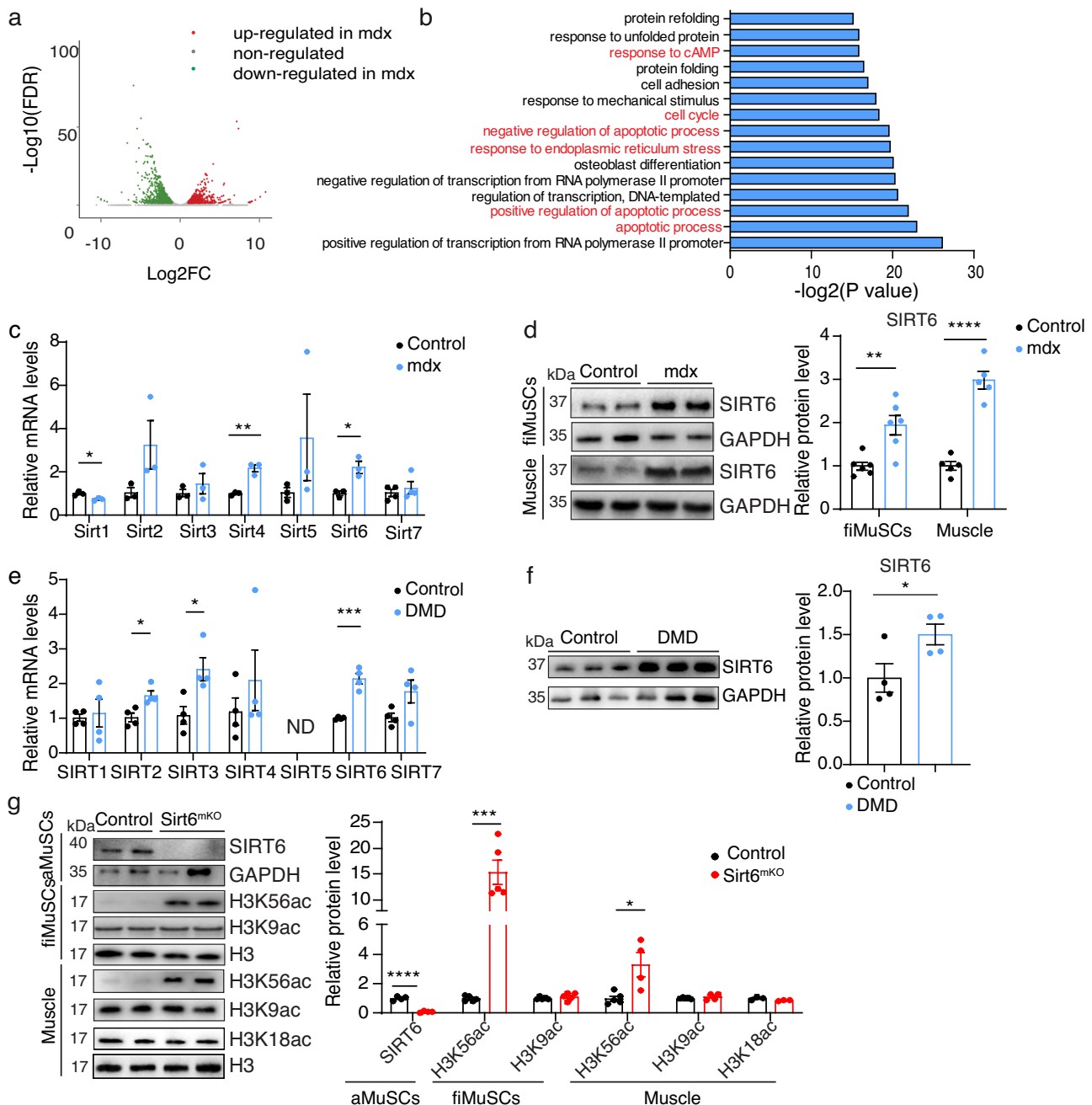

**Fig. 1 SIRT6 is upregulated in muscle cells of *mdx* mice and controls H3K56ac levels. a** Volcano plot of differential expressed genes in freshly isolated MuSCs from wild-type and *mdx* mice. The vertical axis (*y*-axis) corresponds to the mean expression value of log10 (FDR), and the horizontal axis (*x*-axis) displays the log2-fold change value. Red dots represent upregulated transcripts in *mdx* mice; green dots represent downregulated transcripts. Benjamini-Hochberg test, two-sided. **b** Top 15 biological processes (BP) deregulated in *mdx* versus wild type MuSCs. Fisher's exact test, two-sided. **c** RT-qPCR analysis of sirtuin expression in freshly isolated MuSCs (fiMuSCs) from wild type mice (hereafter referred to as control) and *mdx* mice. *m36b4* was used as a reference gene. (*$p = 0.0235$, **$p = 0.0024$, *$p = 0.0136$; *Sirt1-6*: $n = 3$; *Sirt7*: $n = 4$). **d** Western blot analysis of SIRT6 expression in freshly isolated MuSCs (fiMuSCs) and muscle tissues from control and *mdx* mice. GAPDH was used as loading control. Quantification of protein bands are shown on the right (**$p = 0.0037$, ****$p < 0.0001$; fiMuSCs $n = 6$; muscle $n = 5$). **e** RT-qPCR analysis of expression of different sirtuins in myoblasts derived from healthy (referred to as control) and DMD patients (*$p = 0.0145$, *$p = 0.0192$, ***$p = 0.0003$; $n = 4$). ND non-detectable levels. **f** Western blot analysis of SIRT6 expression in myoblasts derived from control and DMD patients. GAPDH was used as loading control. Quantification of protein bands are shown on the right (*$p = 0.0484$; $n = 4$). **g** Western blot analysis of SIRT6 level in proliferating MuSCs isolated from WT and Sirt6^mKO mice 3 days after in vitro culture (ASCs). H3K56ac, H3K9ac, and H3K18ac levels in freshly isolated MuSCs (fiMuSCs) and in muscle tissues from WT and *Sirt6^mKO* mice are shown as well. GAPDH or Histone H3 was used as loading control. Quantification of protein bands are shown in the right (****$p < 0.0001$; ***$p = 0.0003$; *$p < 0.0167$, Sirt6: $n = 4$; H3K56ac, H3K9ac: $n = 5$; H3K18ac: $n = 3$). For **c**–**g** unpaired two-tailed *t*-test. Males, 8–12 weeks old were used. Data are presented as mean ± SEM (**c**–**g**). Source data are provided in the Source Data file.

ICNCre$^{pos}$). PCR-based genotyping revealed that deletion of Sirt6 was highly efficient in MuSCs of Pax7-ICNCre$^{pos}$Sirt6$^{fl/fl}$ mice (hereafter referred as Sirt6$^{mKO}$) (Supplementary Fig. 2a, b). SIRT6 has been described to deacetylate H3K9, H3K18, and H3K56. However, analysis of the acetylation status revealed that loss of Sirt6 only caused a massive increase of histone H3K56ac levels but no detectable change of histone H3K9ac and H3K18ac, indicating that SIRT6 is the dominant deacetylase for H3K56ac in MuSCs and muscle tissue (Fig. 1g). RNA-seq analysis of Sirt6-deficient MuSC identified 189 moderately upregulated genes, which is surprising given the massive increase of H3K56ac (Supplementary Fig. 2c). The upregulated genes belonged to related to muscle contraction, skeletal muscle tissue and organ development and extracellular matrix (Supplementary Fig. 2d, e). The increase in H3K56ac and the resulting expression changes after inactivation of Sirt6 had no obvious effects on fertility, body and muscle weight, MuSC numbers and skeletal muscle morphology (Supplementary Fig. 2d–g, i, k). Likewise, we observed no differences in skeletal muscle regeneration after CTX-induced muscle injury (Supplementary Fig. 2h, i). To further analyze skeletal muscle regenerative capacity, we subjected Sirt6$^{mKO}$ mice to three successive rounds of CTX-induced muscle injury and analyzed formation of new skeletal muscle fibers already 7 days after the last CTX injection. Even under these stringent conditions, skeletal muscle regeneration was not compromised in Sirt6$^{mKO}$ mice, rather the opposite, since we observed significantly higher TA muscle masses in Sirt6$^{mKO}$ compared to WT mice after regeneration, suggesting that Sirt6 inactivation accelerates rather than slows down muscle regeneration after injury. In contrast to germline Sirt6 knockout mice (Sirt6$^{-/-}$), which show severe metabolic dysfunctions, we did not find evidence for glucose intolerance in Sirt6$^{mKO}$ mice or transcriptional activation of glycolytic genes that are repressed by SIRT6 in other cell types[29] (Supplementary Fig. 2i–n).

**Sirt6 inactivation reduces persistent MuSC activation in mdx muscles.** Since dysfunctions of SIRT6 are associated with major human diseases including cancer, neurodegenerative and heart diseases[30], we wondered whether increased expression of Sirt6 in mdx MuSC might serve a role for coping with cellular stress or other biological processes in muscular dystrophy. Analysis of MuSCs in Sirt6 and mdx compound mutant mice (hereafter referred to as Sirt6$^{mKO}$/mdx mice) (Supplementary Fig. 3a) revealed a striking reduction of PAX7$^{+}$/MYOD$^{+}$ activated MuSCs. The percentage of PAX7/MYOD double positive MuSCs, which is strongly increased in mdx mice, essentially dropped to wild type levels in Sirt6$^{mKO}$/mdx TA muscles (Fig. 2a). Consistently, expression of MyoD was reduced in freshly isolated Sirt6$^{mKO}$/mdx compared to mdx MuSCs (Supplementary Fig. 3b). We also detected reduction of other hallmarks of MuSC activation in Sirt6$^{mKO}$/mdx mice, such as reduced heterochromatin content and enlarged nuclear size[18] (Fig. 2b). Taken together, these findings indicate that inactivation of Sirt6 in the muscle lineage of mdx mice mitigates persistent activation of MuSC.

We next determined the gene expression profiles of freshly isolated MuSCs from control, mdx and double mutant mice by RNA-seq. Bioinformatics analysis identified 977 upregulated genes in mdx compared to WT MuSC. Interestingly, inactivation of Sirt6 in mdx mice resulted in downregulation of 173 out of the 977 genes that were upregulated in mdx MuSCs (Fig. 2c). GO term enrichment analysis revealed that these genes are primarily related to biological processes involved in DNA replication and cell cycle regulation (Fig. 2c). Consistently, in vivo EdU incorporation assays revealed a dramatic reduction of MuSCs proliferation in Sirt6$^{mKO}$/mdx muscles, suggesting that loss of

Sirt6 reduced division of MuSCs in mdx mice (Fig. 2d). Furthermore, cell cycle genes, markers of MuSC activation, and stress response-related genes that were upregulated in mdx MuSCs were partially normalized after loss of Sirt6 (Supplementary Fig. 3c–e). In total, 804 of genes that were upregulated in mdx MuSC were not affected by the absence of Sirt6. In total, 335 genes were upregulated in Sirt6$^{mKO}$/mdx but not in mdx mice compared to controls. Only 9 of these 335 genes were also upregulated in Sirt6$^{mKO}$ MuSCs, suggesting that the remaining 326 genes are not direct targets of SIRT6 but are upregulated due to secondary events, such as mitigation of persistent MuSC activation (Supplementary Fig. 3f, g). This conclusion was supported by analysis of H3K9ac. Unlike the increase in H3K56ac, which was present both in Sirt6$^{mKO}$/mdx and Sirt6$^{mKO}$ MuSCs, western blot analysis indicated a substantially increase of H3K9ac only in Sirt6$^{mKO}$/mdx but not in Sirt6$^{mKO}$ MuSCs (Supplementary Fig. 3h). In addition, we identified that 201 genes that were downregulated in both mdx and Sirt6$^{mKO}$/mdx MuSCs compared to control MuSCs and 346 genes that were down-regulated in Sirt6$^{mKO}$/mdx compared to mdx MuSCs (Fig. 2c and Supplementary Fig. 3g). In summary, the results suggest that inactivation of Sirt6 in mdx MuSCs thwarts several features characteristic for activated MuSC in mdx muscles, including reduced heterochromatin content, increased nuclear size, as well as enhanced expression of proliferation- and stress response-related genes.

**Suppression of Sirt6 eliminates several pathological hallmarks characteristic for mdx mice.** Loss of function mutations in the dystrophin gene lead to several characteristic symptoms in mdx mice that can be attributed to impaired sarcolemma integrity. To investigate whether suppression of SIRT6 activity improves the clinical phenotype of mdx mice, we determined several parameters. We found that inactivation of Sirt6 reduced the elevated body weight and TA muscle weight to tibia length ratios in mdx mice, probably by attenuating reactive hypertrophy, which occurs in mdx mice due to enhanced MuSC activation (Fig. 3a, b). To corroborate this hypothesis, we determined the complete skeletal muscle mass in the body. MRI measurements revealed a significant reduction of muscle and fat volume in Sirt6$^{mKO}$/mdx mice (Fig. 3c and Supplementary Fig. 4a). Furthermore, histological analysis of diaphragm and TA muscles from Sirt6$^{mKO}$/mdx mice indicated a clear reduction of hypertrophy compared to mdx animals (Fig. 3d and Supplementary Fig. 4b). To determine the membrane integrity of myofibers in Sirt6$^{mKO}$/mdx mice, we performed injections of Evan's blue and monitored accumulation of the dye in the diaphragm muscle, which is the most severely affected muscle in mdx mice. Remarkably, virtually no Evan's blue staining was observed in diaphragm muscles from Sirt6$^{mKO}$/mdx mice, while a strong staining was visible in mdx mice due to penetration of the dye through the leaky sarcolemma (Fig. 3e and Supplementary Fig. 4c). Leakage of myofibers in mdx mice causes a strong increase of serum creatine kinase (CK) levels, which is proportional to skeletal muscle damage. As expected, we observed a strong decrease of serum CK levels in Sirt6$^{mKO}$/mdx compared to mdx mice, further proving that suppression of SIRT6 activity in skeletal muscles increases the stability of the sarcolemma in mdx mutants (Fig. 3f). Finally, we monitored the motility of control, mdx and Sirt6$^{mKO}$/mdx mice in Phenomaster metabolic cages. Lack of Sirt6 increased the physical activity of mdx mice, while no changes in food /liquid intake and respiratory exchange ratio (RER) were observed (Fig. 3g and Supplementary Fig. 4d–f). Taken together, the data clearly indicate that suppression of SIRT6 attenuates several pathological hallmarks characteristic for mdx mice.

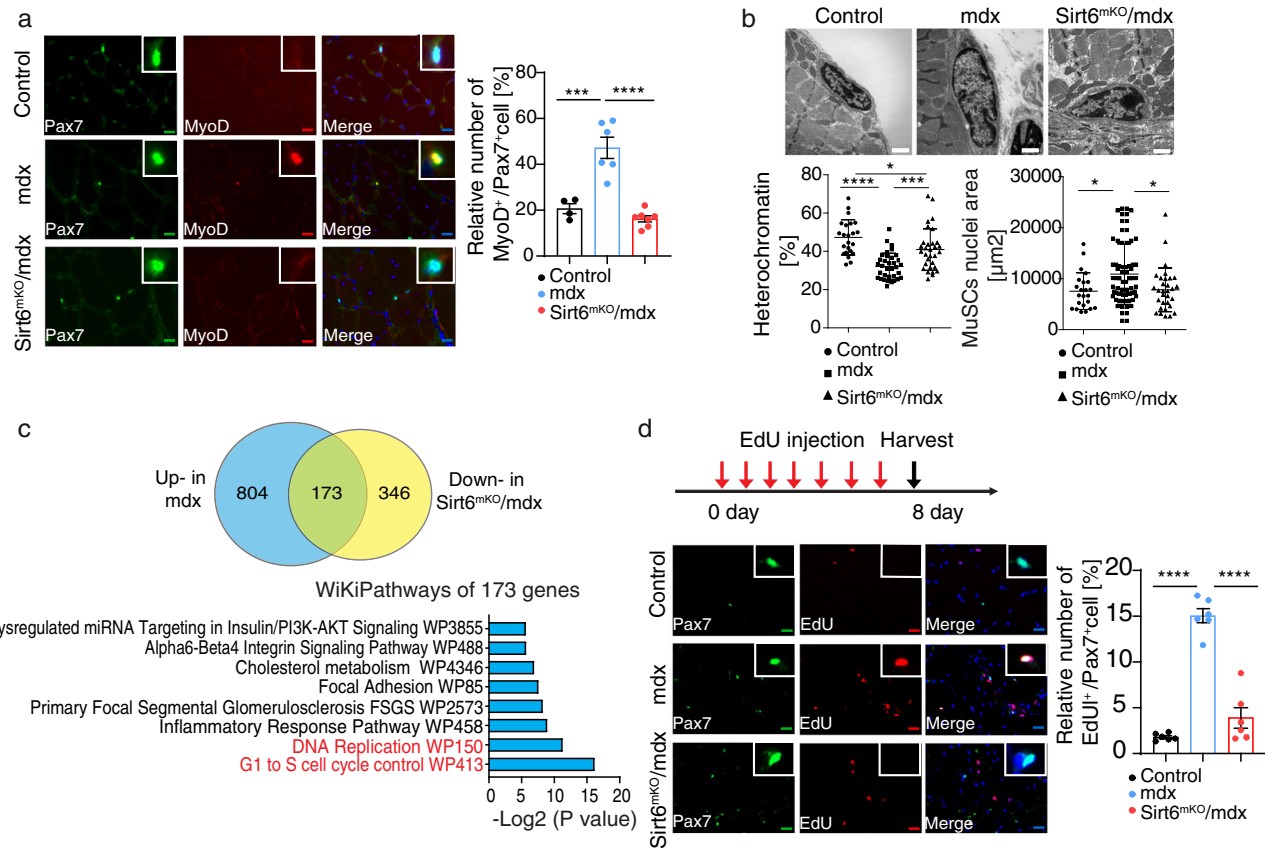

**Fig. 2 Deletion of *Sirt6* reduces persistent activation of MuSC in skeletal muscles of *mdx* mice. a** Immunofluorescence staining of MYOD+/PAX7+ double positive MuSCs in TA muscles of control ($n = 4$), *mdx* ($n = 6$) and *Sirt6mKO/mdx* ($n = 7$) mice. Scale bar: 20 µm. The ratios of MYOD+/PAX7 double positive MuSCs are shown in the right panel (one-way ANOVA with Bonferroni multiple comparisons test: ***$p = 0.002$, ****$p < 0.0001$). **b** EM images of increased heterochromatin content and decreased nuclei size in MuSCs of *Sirt6mKO/mdx* mice. Scale bar: 2 µm. Recordings were quantified with ImageJ software and are shown in the lower panels (one-way ANOVA with Bonferroni multiple comparisons test: heterochromatin content ****$p < 0.0001$, ***$p = 0.0004$; *$p = 0.0379$; nuclei size: *$p = 0.0374$, *$p = 0.0361$, control: $n = 23$; *mdx*: $n = 35$; *Sirt6mKO/mdx*: $n = 31$). **c** Venn diagram of upregulated genes in *mdx* MuSCs compared with control MuSCs and downregulated genes in *Sirt6mKO/mdx* MuSCs compared to *mdx* MuSCs (upper panel) based on RNA-seq analysis. GO term analysis of WikiPathways enriched in overlapping set of 173 genes based on *p* values (lower panel) using Enrichr.
**d** Immunofluorescence staining of EdU+/PAX7+ MuSCs in TA muscles of control, *mdx* and *Sirt6mKO/mdx* mice. Scale bar: 20 µm. A schematic outline of the experimental design is shown in the upper panel. Quantification of EdU+/PAX7+ MuSCs is shown on the right (one-way ANOVA with Bonferroni multiple comparisons test: ****$p < 0.0001$, $n = 6$). Twelve to 16 weeks old male mice were used. Data are presented as mean ± SEM (**a**, **b**, **d**). Source data are provided in the Source Data file.

**Inactivation of Sirt6 partially reverts gene expression patterns of muscle dystrophy related genes in mdx muscles**. Since inactivation of *Sirt6* in *mdx* muscles resulted in improvement of the *mdx* phenotype at the functional level, we wanted to learn more about the underlying molecular mechanisms. Therefore, we performed RNA-seq of muscles from control, *mdx* and *Sirt6mKO/mdx* mice. We identified 4082 differentially expressed genes (DEGs) in *mdx* compared to wild-type muscle and 1597 DEGs in *Sirt6mKO/mdx* compared to *mdx* muscle, indicating that suppression of SIRT6 activity has a strong impact on the course of the disease (Fig. 4a, b). Similar to the RNA-seq analysis of MuSCs, a substantial number of DEGs in *mdx* muscles approached control levels in *Sirt6mKO/mdx* muscle (Fig. 4c, d). Bioinformatics analysis revealed that the 257 genes, which were upregulated in *mdx* and returned to WT levels in *Sirt6mKO/mdx* muscles, are associated with various regulatory pathways including complement cascades (part of innate immune response), cell cycle control, oxidative stress and damage (Fig. 4c). Several of the 266 genes that were downregulated in *mdx* muscles but reached WT levels after *Sirt6* inactivation are involved in different metabolic pathways, including fatty acid biosynthesis and fatty acid beta

oxidation (Fig. 4d). It seems likely that improved membrane stabilization in *Sirt6mKO/mdx* compared to *mdx* muscles averts metabolic changes that occur due to myofiber damage and mitochondrial dysfunction in dystrophic in *mdx* muscles. Along the same line, expression of numerous genes involved in cellular stress response such as *Dab2*, *Hspa2* and *Ptpn22*, which are upregulated in mdx muscles, were reduced after *Sirt6* inactivation, suggesting improvement of cellular physiology in *Sirt6mKO/mdx* mice (Fig. 4e, f and Supplementary Table 2 and Supplementary Fig. 3d). We concluded that inactivation of *Sirt6* facilitates expression of genes that either compensate for dystrophin or prevent activation of disease related genes, thereby ameliorating the *mdx* phenotype.

**Loss of SIRT6 activates the intronic enhancer of utrophin and increases Utrn expression**. Our results identified SIRT6 acts as the main H3K56ac histone deacetylase in skeletal muscle. Although the role of H3K56ac in gene regulation has been studied intensively in yeast, its function in mammalian cells is less well characterized[31]. To obtain insights into the genome-wide distribution of H3K56ac in quiescent MuSCs, we performed ChIP-seq experiments. We

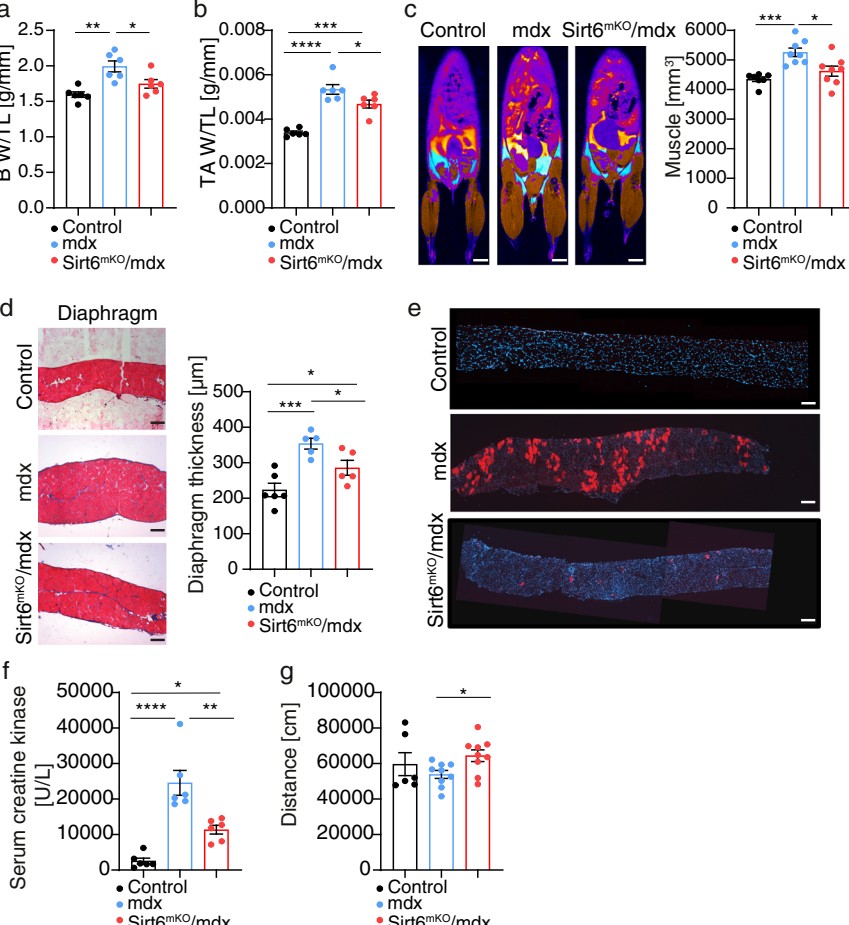

**Fig. 3 Absence of *Sirt6* improves membrane integrity of dystrophic muscle fibers. a** Body weight/tibia length ratios of control, *mdx* and *Sirt6^{mKO}/mdx* mice (one-way ANOVA with Bonferroni multiple comparisons test: *$p = 0.0354$, **$p = 0.001$, $n = 6$ for each group). **b** TA muscle weight/tibia length ratios of control, *mdx* and *Sirt6^{mKO}/mdx* mice (one-way ANOVA with Bonferroni multiple comparisons test: *$p = 0.0388$, ***$p = 0.002$, ****$p < 0.0001$, $n = 6$ for each group). **c** MRI analysis of muscle and fat mass of 12-week-old control ($n = 7$), *mdx* ($n = 8$) and *Sirt6^{mKO}/mdx* ($n = 8$) littermates. Scale bar: 5 mm. Quantification of muscle volume are shown on the right (one-way ANOVA with Bonferroni multiple comparisons test: *$p = 0.0108$, ***$p = 0.0006$). **d** Masson's trichrome staining of the diaphragm muscle in control ($n = 6$), *mdx* ($n = 5$) and *Sirt6^{mKO}/mdx* ($n = 5$) mice (one-way ANOVA with Benjamini and Yekuteili multiple comparisons test: *$p = 0.0263$, *$p = 0.0320$, ***$p = 0.0002$). Scale bar: 100 µm. **e** Evans blue staining of diaphragm muscles from control, *mdx* and *Sirt6^{mKO}/mdx* mice ($n = 4$ for each group). Cross-sectional images of decreased myofiber leakage in *Sirt6^{mKO}/mdx* mice are shown. Scale bar: 100 µm. **f** Serum creatine kinase (CK) enzyme activity in control, *mdx* and *Sirt6^{mKO}/mdx* mice (one-way ANOVA with Bonferroni multiple comparisons test: *$p = 0.0380$, **$p = 0.0022$, ****$p < 0.0001$; $n = 6$ for each group). **g** Physical activity of control ($n = 6$), *mdx* ($n = 9$) and *Sirt6^{mKO}/mdx* ($n = 9$) mice in metabolic cages (PhenoMaster) (one-way ANOVA with Benjamini and Yekuteili multiple comparisons test: *$p = 0.0486$). Twelve to 16 weeks old male mice were used. Data are presented as mean ± SEM (**a–d**, **f**, **g**). Source data are provided in the Source Data file.

identified 15,237 H3K56ac peaks that overlapped with H3K4me3 (33%), H3K27ac (8.8%) or both modifications (37%), suggesting that H3K56ac is enriched at active promoters and enhancers (Fig. 5a). The number of H3K56ac peaks was globally increased in *Sirt6^{mKO}* MuSCs, matching the higher amount of H3K56ac detected by western blot analysis (Fig. 5b). Notably, around 46% of newly emerging H3K56ac peaks in *Sirt6^{mKO}* MuSCs localize within distal intergenic regions, which corresponds to 10–100 kb up- or downstream of TSS, regions that often contain enhancers (Fig. 5c, d). Only 13% of upregulated H3K56ac peaks were found within promoters (Fig. 5c). Furthermore, we compared H3K56ac peaks within promoters and enhancers of freshly isolated MuSCs to published H3K4me3 and H3K27ac ChIP-seq datasets, assuming that overlapping peaks of H3K4me3 and H3K27ac (H3K4me3+/H3K27ac+) represent promoters. Since no ChIP-seq data for enhancer markers such H3K4me1 or p300 are available for freshly isolated MuSCs, we defined H3K4me3 negative but H3K27ac positive regions (H3K4me3−/H3K27ac+) as enhancers. In line with the results

shown above, upregulated H3K56ac peaks in *Sirt6^{mKO}* were mostly located at enhancers (H3K4me3−/H3K27ac+) but not at promoters (H3K4me3+/H3K27ac+) (Fig. 5e), suggesting that SIRT6 specifically modulates enhancer activity. Similarly, we found globally increased H3K56ac peaks and a substantial overlap of differentially increased H3K56ac peaks with enhancers marked by H3K27ac and H3K4me1 in *Sirt6* knockdown mouse embryonic stem cells (*Sirt6^{KD}* mESC), suggesting that modulation of enhancer activity by H3K56ac deacetylation is a common function of SIRT6 in various cell types (Supplementary Fig. 5a–e). To investigate the impact of SIRT6-mediated H3K56 deacetylation on chromatin accessibility in *Sirt6^{mKO}* MuSCs, we overlapped upregulated H3K56ac peaks with differential accessible chromatin regions, identified by ATAC-seq. We identified 1593 genes that contain both upregulated H3K56ac peaks and show enhanced chromatin accessibility (Fig. 5f).

Intriguingly, we detected utrophin (*Utrn*) within the group of genes with upregulated H3K56ac peaks and increased chromatin accessibility. UTRN is a paralogue of dystrophin and can

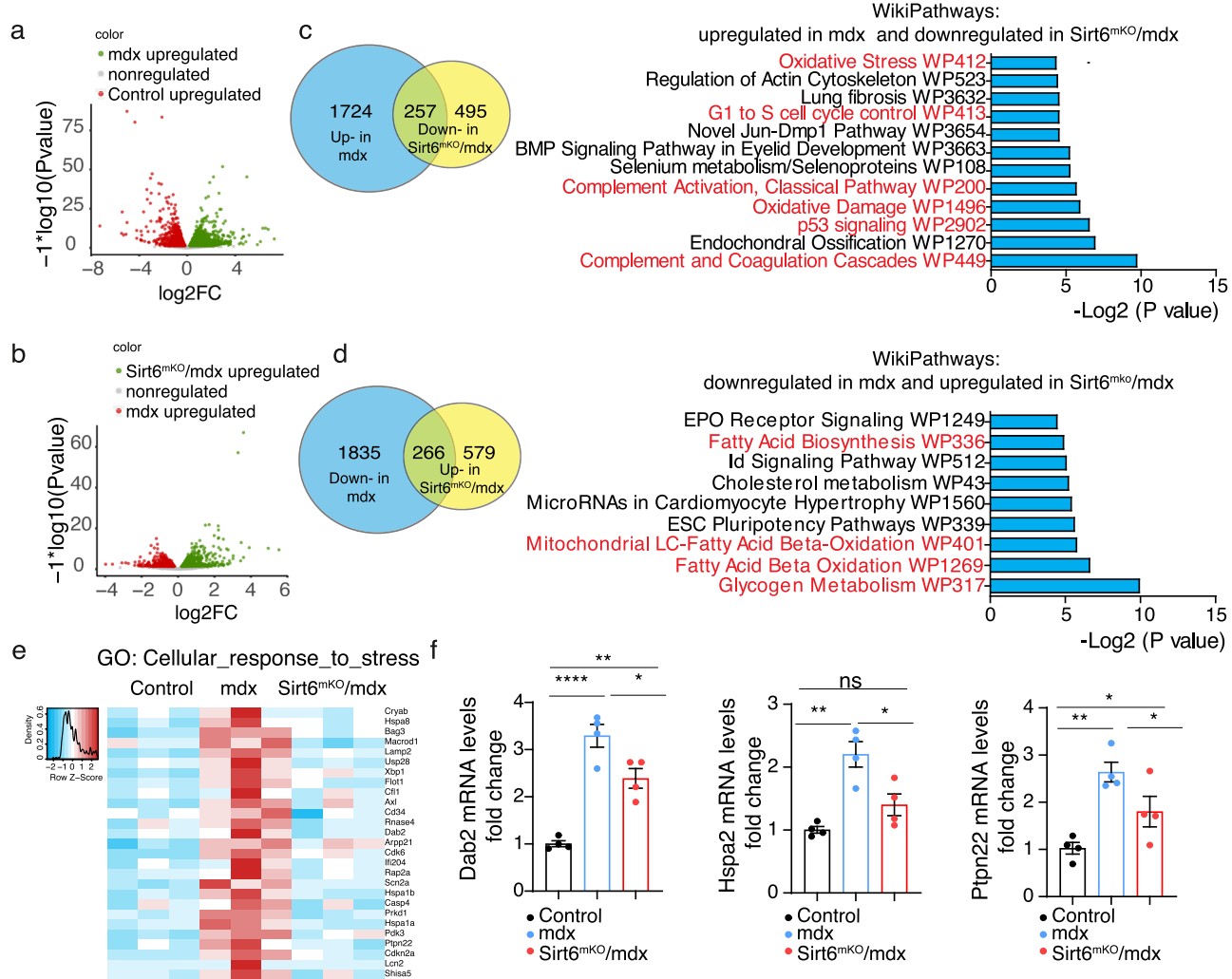

**Fig. 4 Loss of *Sirt6* partially normalizes aberrant expression patterns in dystrophic *mdx* muscles. a** Volcano plot of differentially expressed genes in *mdx* compared to control muscles. Benjamini-Hochberg test, two-sided. **b** Volcano plot of differentially expressed genes in *mdx* compared to *Sirt6^mKO^/mdx* muscles. Benjamini-Hochberg test, two-sided. **c** Venn diagram of overlapping upregulated genes in *mdx* muscle compared to control muscle and downregulated genes in *Sirt6^mKO^/mdx* muscle compared to *mdx* muscles based on RNA-seq. In total, 257 overlapping genes were categorized by Wikipathways analysis. Fisher's exact test, two-sided. **d** Venn diagram of overlapping downregulated genes in *mdx* compared to control muscle and upregulated genes in *Sirt6^mKO^/mdx* muscle compared to *mdx* muscle based on RNA-seq. In total, 266 overlapping genes were categorized by Wikipathways analysis. Fisher's exact test, two-sided. **e** Heat map of expression of genes involved in cellular stress responses in control, *mdx* and *Sirt6^mKO^/mdx* muscle (n = 3 for each group). **f** RT-qPCR analysis of expression of indicated genes involved in cellular stress responses in control, *mdx* and *Sirt6^mKO^/mdx* muscles. *m36b4* was used as reference gene (one-way ANOVA with Bonferroni multiple comparisons test: *Dab2:* *$p = 0.0235$, **$p = 0.0017$; ****$p < 0.0001$; *Hspa2:* ns not significant, *$p = 0.0164$, **$p = 0.0012$; *Ptpn22:* **$p = 0.009$, *$p = 0.0316$, *$p = 0.0430$; n = 4 for each group). Twelve to 16 weeks old male mice were used. Data are presented as mean ± SEM. Source data are provided in the Source Data file.

partially compensate for the loss of dystrophin in skeletal muscles[32]. We found that UTRN protein level was higher in *Sirt6^mKO^* muscle compared to WT muscle (Supplementary Fig. 6a), confirming that SIRT6 represses *Utrn* gene expression. Even more interestingly, RT-qPCR revealed a substantial increase of *Utrn* expression in *Sirt6^mKO^/mdx* muscles compared to *mdx* muscles (Fig. 5g). Consistent with previous studies[33], *Utrn* expression was increased in *mdx* compared to control muscles (Fig. 5g). In contrast, no increase of *Utrn* expression was detected in freshly isolated MuSCs, suggesting that chromatin changes induced by the absence of *Sirt6* precede transcriptional changes in myofibers (Supplementary Fig. 6b, c). We reasoned that important factors required for expression of *Utrn* are missing in MuSCs, which also corresponds to very low mRNA levels of *Utrn* mRNA in MuSC. Immunofluorescence staining and western

blot analysis confirmed a further increase of URTN in *Sirt6^mKO^/mdx* compared to *mdx* muscles, although *mdx* muscles already have higher levels of URTN compared to WT muscles (Fig. 5h, i).

Closer inspection of the upregulated H3K56ac peaks after loss of *Sirt6* guided us to the first intron, which contains several cis-regulatory elements including a putative mouse DUE located in 3.5 kb downstream of the commonly used *Utrn* TSS (Fig. 5j). In contrast, no upregulated H3K56ac peak was detected in the vicinity of *Utrn* proximal promoter regions (H3K4me3+/H3K27ac+) (Supplementary Fig. 6d). The putative mouse DUE shows 84% identity with the human 128 bp DUE and contains a consensus binding site for AP1 (Supplementary Fig. 6e). ChIP-qPCR experiments performed in myotubes derived from C2C12 muscle cells demonstrated that SIRT6, P300 and AP1 bind the DUE of the *Utrn* gene (Fig. 5k). Similar results were obtained in ChIP-qPCR

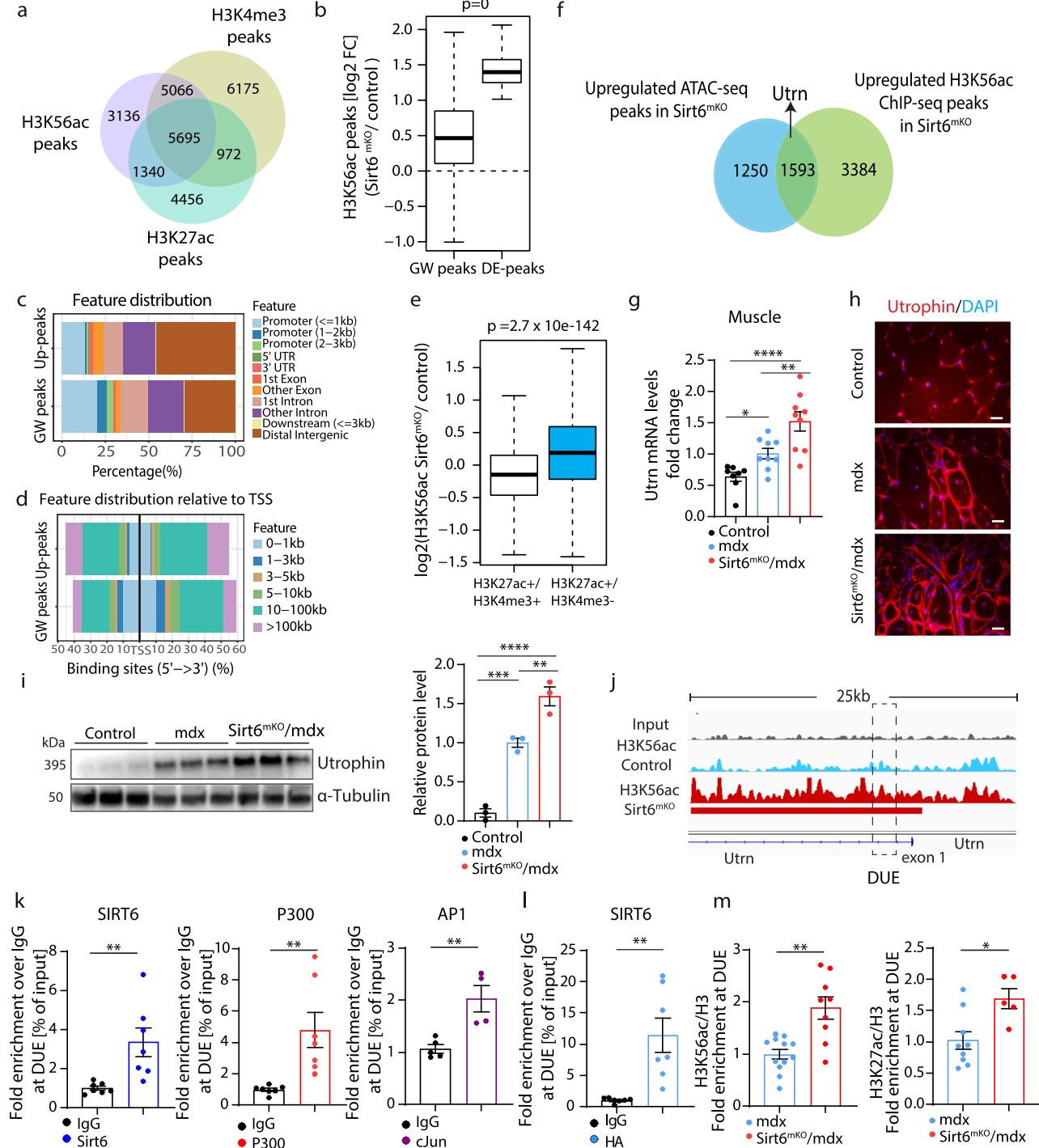

experiments for SIRT6 when primary myotubes expressing lentivirally-transduced, HA-tagged SIRT6 were employed (Fig. 5l). ChIP-qPCR for H3K56ac and H3K27ac confirmed a significant increase of H3K56ac at the DUE as well as an increase of the enhancer mark H3K27ac in *Sirt6^mKO^/mdx* compared to *mdx* muscles (Fig. 5m). The results clearly demonstrate that SIRT6 directly deacetylates H3K56 at the DUE of the *Utrn* gene, suppressing *Utrn* expression.

In addition to *Utrn*, we found a strong increase of *Mstn* expression in *Sirt6^mKO^/mdx* compared to *mdx* and control muscles. MSTN is a negative regulator of muscle mass and a known downstream target of SIRT6[34]. Expression of *Mstn* was

strongly reduced in *mdx* compared to control muscles, which is probably part of a compensatory circuit leading to hypertrophy in *mdx* versus control myofibers (Supplementary Fig. 7a–c). In line with these observations, we detected higher H3K56ac and H3K27ac levels at a region that was previously defined as the enhancer/promoter of the *Mstn* gene[35], when comparing *Sirt6^mKO^/mdx* to *mdx* (Supplementary Fig. 7d). The increased deposition of H3K56ac deposition within the enhancer/promoter of *Mstn* gene was coupled with hypo-methylation of the regulatory element located +264 bp downstream of TSS (Supplementary Fig. 7e), a locus that is negatively correlated with *Mstn* expression when hyper-methylated[36]. Taken together, we

**Fig. 5 SIRT6 controls H3K56ac deposition and enhancer activity of the *Utrn* gene. a** Venn diagram of H3K56ac, H3K27ac and H3K4me3 ChIP-seq data in wild type fiMuSCs. **b** Boxplot of upregulated H3K56ac peaks in fiMuSCs of *Sirt6*$^{mKO}$ mice. Y-axis: log2-fold changes (Log2FC), GW: all Genome-Wide peaks, DE: significantly deregulated peaks ($n = 2$). **c**, **d** Distribution of H3K56ac ChIP-seq peaks across genomic regions (**c**) and relative to TSS (**d**). **e** Boxplot demonstrating that the increase of H3K56ac peaks in *Sirt6*$^{mKO}$ is higher across enhancer-associated H3K56ac peaks (positive for H3K27ac but negative for H3K4me3) compared to promoter H3K56ac peaks (positive for H3K4me3) ($n = 2$). **f** Venn diagram of overlapping upregulated ATAC-seq and H3K56ac ChIP-seq peaks *Sirt6*-deficient fiMuSCs. **g** RT-qPCR analysis of *Utrn* expression in muscles from control ($n = 8$), *mdx* ($n = 9$) and *Sirt6*$^{mKO}$/*mdx* ($n = 9$) mice. Reference gene: *m36b4*. (One-way ANOVA with Benjamini multiple comparisons test: *$p = 0.0304$, **$p = 0.003$ ****$p < 0.0001$). **h** Immunofluorescence staining for UTRN on TA sections from control, *mdx* and *Sirt6*$^{mKO}$/*mdx* mice. Scale bar: 20 µm. $n = 3$ per group. **i** Western blot analysis of utrophin expression in muscles from control, *mdx* and *Sirt6*$^{mKO}$/*mdx* mice. Loading control: α-tubulin. Quantification is on the right. (One-way ANOVA with Bonferroni multiple comparisons test: **$p = 0.070$, ***$p = 0.0008$, ****$p < 0.0001$; $n = 3$ per group). **j** H3K56ac distribution in the proximal promoter region and DUE of the *Utrn* gene showing differential H3K56ac peaks in control and *Sirt6*$^{mKO}$ MuSCs. **k** ChIP-qPCR analysis of Sirt6 (IgG: $n = 7$; SIRT: $n = 7$), p300 (IgG: $n = 7$; p300: $n = 7$) and cJun (IgG: $n = 5$; cJun: $n = 4$) at the DUE of the *Utrn* gene in C2C12 myotubes. (**$p = 0.0074$, **$p = 0.0054$, **$p = 0.0055$). **l** ChIP-qPCR analysis of SIRT6 at the *Utrn* DUE in HA-SIRT6 overexpressing MuSCs derived myotubes (**$p = 0.0026$, $n = 7$ per condition). **m** ChIP-qPCR analysis of H3K56ac and H3K27ac at the DUE of the *Utrn* gene in *mdx* (H3K56ac: $n = 12$; H3K27ac: $n = 9$) and *Sirt6*$^{mKO}$/*mdx* (H3K56ac: $n = 9$; H3K27ac: $n = 5$) muscles (***$p = 0.005$, *$p = 0.0126$). Data are presented as mean ± SEM (**g**, **i**, **k**–**m**). Box plots in **b**, **e** were defined by default in R, 1.5 × IQR (interquartile range). The whiskers show minimum and maximum values. Wilcoxon rank sum test, two-sided $p < 2.2 × 10^{16}$. For **k**–**m** unpaired two-tailed *t*-test. Twelve to 16 weeks old male mice were used. Source data are provided in the Source Data file.

hypothesize that augmented expression of *Utrn* and *Mstn* in *Sirt6*$^{mKO}$/*mdx* muscle might contribute to the improvement of muscular dystrophy in *mdx* mice.

**Targeted deacetylation of H3K56ac at the intronic enhancer by SIRT6 represses *Utrn* transcription.** The increase of *Utrn* expression and H3K56 acetylation at the DUE of *Utrn* after loss of *Sirt6* argues for the repression of *Utrn* expression by SIRT6 but does not provide direct proof for such a mechanism. Thus, we decided to manipulate H3K56 acetylation within the DUE by specific recruitment of SIRT6 via a CRISPR-dCas9 approach[37]. We expressed different fusion proteins consisting of catalytically inactive dCas9 and either active or inactive versions of SIRT6 (dCas9-Sirt6wt or dCas9-Sirt6mut) together with control or sequence-specific gRNAs targeting the DUE in MuSCs using a lentivirus system (Fig. 6a). Since the gRNA lentiviral constructs contained a mCherry reporter and the dCas9-Sirt6/p300 lentivirus a BFP reporter, we were able to monitor co-transduction efficiencies. Three days after initiation of differentiation, more than 90% of differentiated myotubes were mCherry$^+$ (Fig. 6b). Recruitment of SIRT6 to the DUE resulted in a decline of H3K56ac levels in *Sirt6*$^{mKO}$ myotubes (Fig. 6c). Importantly, *Utrn* expression was reduced after recruitment of the enzymatically active but not the catalytically dead version of SIRT6 to the DUE (Fig. 6d). To analyze the functional consequences of targeted H3K56ac deacetylation at the intronic enhancers for *Utrn* expression, we performed similar experiments in *Sirt6*$^{mKO}$/*mdx* myotubes. We found that recruitment of wildtype but not catalytically inactive SIRT6 to the DUE resulted in a decline of H3H56ac levels and also reduced *Utrn* expression (Fig. 6e, f). Furthermore, we stressed membranes of MuSC-derived myotubes treatment by a hypoosmotic shock and measured CK leakage after co-transduction of dCas9-SIRT6 with control or DUE-specific gRNAs. We found that dCas9-mediated recruitment of active but not inactive SIRT6 to the DUE increased CK levels in supernatants of myotubes derived from *Sirt6*$^{mKO}$/*mdx* MuSCs (Fig. 6g).

In addition, we used a similar approach in wild-type MuSCs to recruit either active or inactive versions of the histone acetyltransferase p300 to the DUE of *Utrn*. Recruitment of dCas9-p300wt to the *Utrn* gene increased H3K56ac levels at the DUE and augmented *Utrn* expression in wild-type MuSC-derived myotubes, which normally show low levels of H3K56ac (Fig. 6h, i). Of note, expression of dCas9-p300mut resulted in a minor decrease of *Utrn* expression, indicating a dominant negative effect after recruitment of mutant p300 to the DUE (Fig. 6i). Recruitment of WT p300 to

the DUE reduced the concentration of CK in supernatants of *mdx* myotubes, while no changes were present when an inactive form of p300mut was used (Fig. 6j), exactly opposite to the results obtained by recruitment of active SIRT6 in *Sirt6*$^{mKO}$/*mdx* myotubes, which resulted in increased leakage of CK.

**UTRN is instrumental for SIRT6-dependent amelioration of muscular dystrophy in mdx mice.** To proof that upregulation of utrophin plays a major role for attenuation of the dystrophic pathology in *Sirt6*$^{mKO}$/*mdx*, we inactivated the *Utrn* gene in *Sirt6*$^{mKO}$/*mdx* mutant mice. MRI measurements of control, *Utrn*$^{−/−}$/*mdx* and *Sirt6*$^{mKO}$/*Utrn*$^{−/−}$/*mdx* mice revealed no significant reduction of muscle but a strong decrease of fat volume in both *Utrn*$^{−/−}$/*mdx* and *Sirt6*$^{mKO}$/*Utrn*$^{−/−}$ mice (Supplementary Fig. 8a). TA weight/tibia length ratios of *Sirt6*$^{mKO}$/*Utrn*$^{−/−}$/*mdx* even increased slightly compared to control mice (Supplementary Fig. 8b), No significant differences were detected among control, *Utrn*$^{−/−}$/*mdx* and *Sirt6*$^{mKO}$/*Utrn*$^{−/−}$/*mdx* mice when body weight was normalized to tibia length (Supplementary Fig. 8b). suggesting that the combined loss of *Utrn* and dystrophin attenuates body growth, which is not compensated for by inactivation of *Sirt6*. H&E staining of TA and diaphragm muscles revealed massive myofiber degeneration and smaller myofibers in diaphragm but not in limb muscles of *Utrn*$^{−/−}$/*mdx* and *Sirt6*$^{mKO}$/*Utrn*$^{−/−}$/*mdx* mice, indicating that absence of UTRN prevents the beneficial effects of SIRT6 on muscle dystrophy in *mdx* mice (Supplementary Fig. 8d). In line with these findings, the absence of *Utrn* prevented reduction of serum CK levels in *Sirt6*$^{mKO}$/*mdx* mutants (Fig. 7a), suggesting that the compromised integrity of the sarcolemma was not restored by SIRT6 when UTRN is absent. To corroborate this finding, we performed Evan's blue injections and monitored uptake of the dye into the diaphragm muscle. Inactivation of the *Utrn* gene in *mdx* mice resulted in enhanced uptake of Evan's blue, indicating that the combined absence of *Utrn* and dystrophin further destabilizes the sarcolemma. Importantly, deletion of *Sirt6* in *Utrn*$^{−/−}$/*mdx* mutant mice did not reduce Evan's blue uptake demonstrating that upregulation of *Utrn* due to the absence of SIRT6-mediated repression is critical for improvement of the muscular dystrophy phenotype in *mdx* mice (Fig. 7b). Furthermore, we observed that deletion of *Sirt6* did not improve survival of *Utrn*$^{−/−}$/*mdx* mice, which die prematurely before 10 weeks of age due to the aggravated phenotype (Fig. 7c). Likewise, deletion of *Sirt6* did not reduce the increased expression of various stress response genes such as *Dab2*, *Hspa2* and *Ptpn22* in *Utrn*/*mdx* mutants (Fig. 7d).

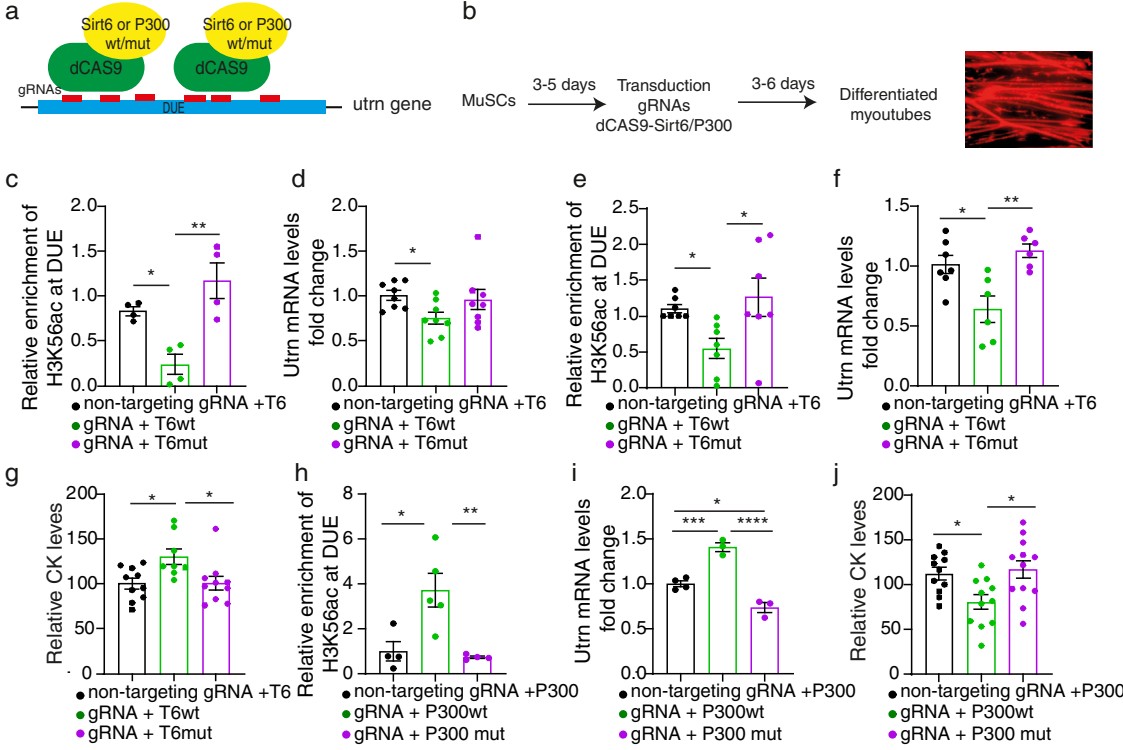

**Fig. 6 CRISPR-dCas9/gRNA-dependent recruitment of SIRT6 and p3OO to the DUE controls *Utrn* expression in muscle cells. a** Recruitment of catalytically active or inactive SIRT6 or P300 to the DUE via CRISPR-dCAS9/gRNA. **b** Experimental design for lentivirus transduction of MuSCs and in vitro differentiation. **c, d** ChIP-qPCR of H3K56ac at the DUE (**c**, $n = 4$ for each group) and RT-qPCR analysis of *Utrn* expression (**d**, $n = 8$ for each group) in *Sirt6*$^{mKO}$ myotubes after co-transduction with non-targeting gRNA or all six DUE-targeting gRNAs and SIRT6wt-dCas9 (T6wt) or SIRT6mut-dCas9 (T6mut). A non-targeted region was used for normalization (one-way ANOVA with Bonferroni multiple comparisons test: **c**: *$p = 0.0368$, **$p = 0.0025$; **d**: *$p = 0.0384$). **e, f** ChIP-qPCR of H3K56ac at the DUE (**e**, $n = 7$ for each group) and RT-qPCR analysis of *Utrn* expression (**f**, non-targeting gRNA: $n = 7$; T6wt/T6mut: $n = 6$) in *Sirt6*$^{mKO}$/*mdx* myotubes after co-transduction with non-targeting gRNA or all six gRNAs targeting DUE of utrophin gene and SIRT6wt-dCas9 or SIRT6mut-dCas9 (one-way ANOVA with Benjamini multiple comparisons test: **e**: *$p = 0.0410$; *$p = 0.0111$; **f**: *$p = 0.0165$, **$p = 0.0029$). **g** CK activity of osmotically stressed derived *Sirt6*$^{mKO}$/*mdx* myotubes after co-transduction with non-targeting gRNA ($n = 10$) or all six DUE-targeting gRNAs and SIRT6wt-dCas9 ($n = 8$) or SIRT6mut-dCas9 ($n = 10$) (One-way ANOVA test with Bonferroni multiple comparisons test: *$p = 0.0259$, *$p = 0.0306$). **h** ChIP-qPCR of H3K56ac levels at the DUE in WT myotubes after co-transduction with non-targeting gRNA ($n = 4$) or all six DUE-targeting gRNAs and P300wt-dCas9 ($n = 5$) or P300mut-dCas9 ($n = 4$) (One-way ANOVA test with Bonferroni multiple comparisons test: *$p = 0.0172$; **$p = 0.0098$). **i** RT-qPCR analysis of *Utrn* expression in WT myotubes after co-transduction with non-targeting gRNA ($n = 4$) or all six DUE-targeting gRNAs and P300wt-dCas9 ($n = 3$) or P300mut-dCas9 ($n = 3$) (one-way ANOVA test with Bonferroni multiple comparisons test: *$p = 0.0104$; ***$p = 0.009$, ****$p < 0.0001$). **j** CK activity of stressed myotubes from *mdx* MuSCs after co-transduction with all six DUE-targeting gRNAs and P300wt-dCas9 ($n = 11$) or P300mut-dCas9 ($n = 12$). CK levels were normalized using non-targeting gRNA ($n = 11$) (One-way ANOVA test with Bonferroni multiple comparisons test: *$p = 0.0388$, *$p = 0.0123$). Twelve to 22 weeks old male and female mice were used. Data are presented as mean ± SEM (**c–j**). Source data are provided in the Source Data file.

## Discussion

In this study, we demonstrated that SIRT6 is the dominant H3K56ac deacetylase in muscle cells but has minor effects on the overall levels of H3K9ac and H3K18ac. Increased levels of H3K56ac in muscle-specific *Sirt6* mutants resulted in increased expression of *Utrn*, which compensates to a certain degree for the absence of dystrophin in *mdx* mice, thereby reducing damage of myofibers and partially ameliorating muscular dystrophy. SIRT6 is assumed to act mainly as a transcriptional repressor by deacetylation of H3K9ac, H3K18ac and H3K56ac[38–40]. We observed massively enhanced concentrations of H3K56ac but not of H3K9ac and H3K18ac in *Sirt6*-deficient muscles. Apparently, other histone deacetylases are able to compensate for the absence of *Sirt6* in maintaining regular levels of H3K9ac and H3K18ac but not of H3K56ac in muscle nuclei.

The function of H3K56ac has been intensely studied in yeast, where it does not only regulate promoter activity but also plays a role in nucleosome assembly[41]. Previous studies demonstrated that H3K56ac is also present at regulatory regions of actively

transcribed genes in human embryonic stem cells, albeit to a lesser extent than in yeast[31]. Our H3K56ac ChIP-seq data revealed that in muscle cells H3K56ac peaks largely overlap with H3K4me3 and H3K27ac depositions, which mark active promoters and enhancers, respectively. We found that prevention of H3K56ac deacetylation by inactivation of *Sirt6* resulted in much broader peaks than the sharp peaks characteristic for H3K4me3 and H3K27ac and several other histone modifications. Upregulated H3K56ac peaks in *Sirt6* mutant MuSCs were mostly localized within distal intergenic regions and highly correlated with putative enhancers, marked by H3K27ac+/H3K4me3−. We also observed a strong enrichment of H3K56ac peaks at H3K27ac+/H3K4me1+ enhancers in mESCs after knockdown of *Sirt6*, suggesting that SIRT6 modulates enhancer activity not only in muscle but also in other cell types. Similar observations were made in drosophila, demonstrating that H3K56ac peaks are broadly distributed over enhancer regions[42]. From those finding it was proposed that an increase in H3K56 acetylation indicates enhancer activation[42]. However, the mechanism by which

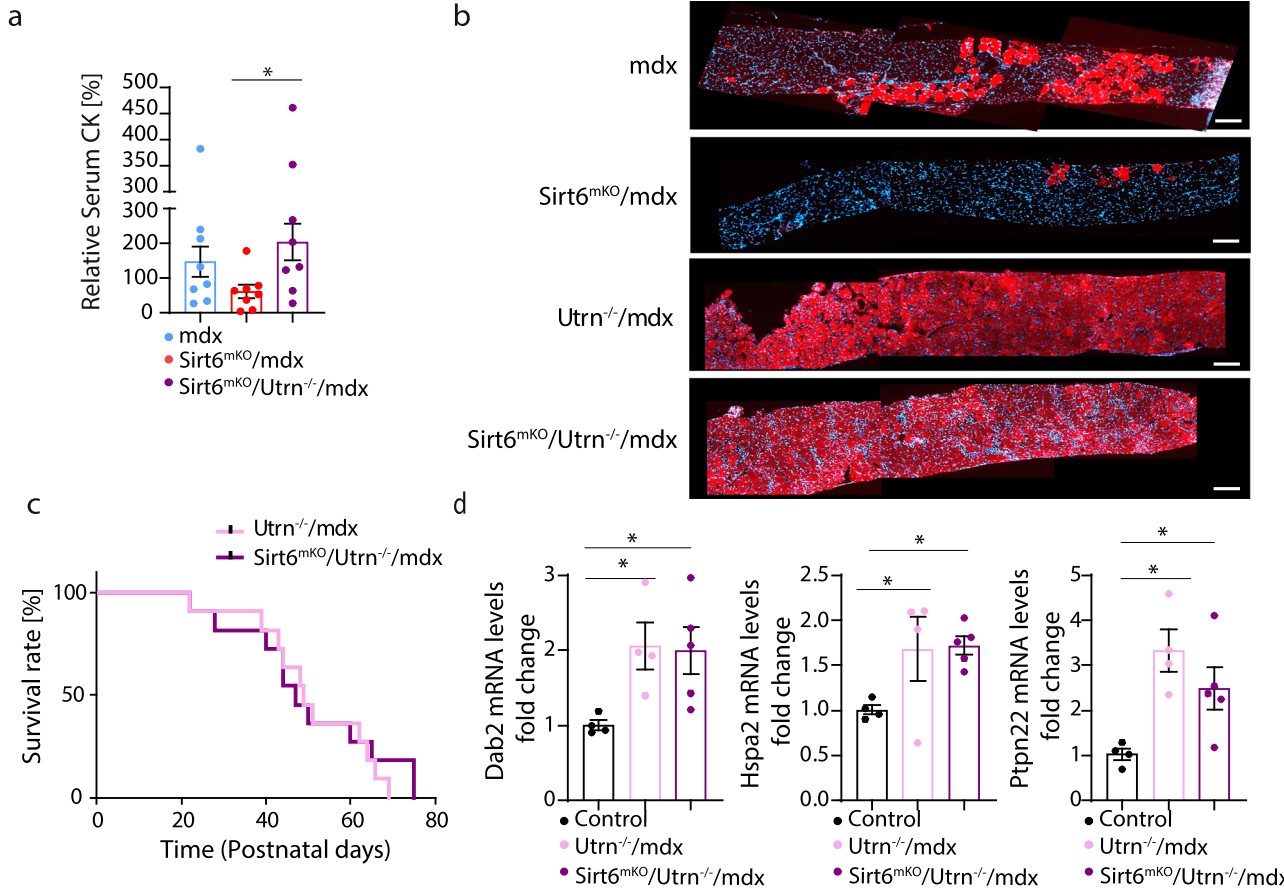

**Fig. 7 Deletion of *Utrn* abrogates beneficial effects of *Sirt6^mKO/mdx* mice. a** Serum creatine kinase (CK) enzyme activity relative to *mdx* (n = 8), *Sirt6^mKO/mdx* (n = 8), and *Sirt6^mKO/Utrn^−/−/mdx* (n = 8) mice (One-way ANOVA with Benjamini multiple comparisons test: *p = 0.0229). **b** Evans blue staining of diaphragm muscles from *mdx* (n = 3), *Sirt6^mKO/mdx* (n = 4), *Utrn^−/−/mdx* (n = 2) and *Sirt6^mKO/Utrn^−/−/mdx* (n = 4) mice. Cross-sectional images showing myofiber leakage in red color are shown. Scale bar: 100 µm. **c** Cumulative survival curve of *Utrn^−/−/mdx* (n = 11) and *Sirt6^mKO/utrn^−/−/mdx* mice (n = 11). Kaplan–Meier curve and Log-rank test. **d** RT-qPCR analysis of expression of genes involved in cellular stress responses in control (n = 4), *Utrn^−/−/mdx* (n = 4), and *Sirt6^mKO/Utrn^−/−/mdx* (n = 5) muscles. *m36b4* was used as a reference gene (One-way ANOVA with Benjamini multiple comparisons test: *Dab2*: *p = 0.0259, *p = 0.0265; *Hspa2*: *p = 0.0437, *p = 0.0280; *Ptpn22*: *p = 0.0034, *p = 0.0290). Five to eight weeks old male and female mice were used. Data are presented as mean ± SEM. Source data are provided in the Source Data file.

H3K56ac stimulates gene activity is incompletely understood. According to a popular hypothesis H3K56ac suppresses integration of linker histones such as H1.0, thereby preventing nucleosome compaction in enhancer regions[43]. Yet, further research seems necessary to fully understand the mechanism by which H3K56ac activates transcription.

We were astonished to see that the massive increase of H3K56ac in *Sirt6*-mutant muscle cells had no obvious effects on muscle development and homeostasis. We did not observe any effects in body and muscle weights, or in muscle morphology in *Sirt6^mKO* compared to WT mice, which is in contrast to germline *Sirt6* knockout mice (*Sirt6^−/−*) that are smaller in size, show severe metabolic dysfunctions including hypoglycemia and loss of subcutaneous fat and display signs of skeletal muscle degeneration[44]. Furthermore, *Sirt6^mKO* mice retained robust regeneration capacity following CTX-induced muscle injury, indicating that the muscle phenotype in *Sirt6^−/−* mice does not originate from within the skeletal muscle lineage but is caused by defects in other cells types, probably due to systemic metabolic changes. This conclusion is also supported by the absence of morphological abnormalities in *MCK-Sirt6* KO mice, lacking *Sirt6* in differentiated myotubes and cardiomyocytes[27]. Surprisingly, we did not find evidence for glucose intolerance and abnormal energy expenditure (EE) in 4-months old *Sirt6^mKO*

mice, which were reported for 7-months old *MCK-Sirt6* KO mice[27]. It is possible that this discrepancy is due to differences in age or the genetic background, the use of a Cre driver strain that already inactivates *Sirt6* very early during muscle development, or the additional deletion of *Sirt6* in the heart in case of *MCK-Sirt6* KO mice. In fact, inactivation of *Sirt6* in cardiomyocytes was reported to cause hypertrophic cardiomyopathy and heart failure, which will have consequences for skeletal muscle physiology[45].

The massive increase of H3K56ac in skeletal muscles due to inactivation of *Sirt6* resulted in de-repression of several genes, including *Utrn* and *Mstn*. By specifically recruiting enzymatically active and inactive versions of SIRT6 to the *Utrn* enhancer via a CRISPR-dCas9/gRNA approach, we demonstrated that the upregulation of *Utrn* in *Sirt6^mKO* mice is not a secondary event, but reflects a direct role of SIRT6 in the regulation of *Utrn* transcription. Targeted recruitment of SIRT6 to the *Utrn* locus in *Sirt6^mKO/mdx* muscle cells also proved that SIRT6-mediated control of *Utrn* expression is decisive to increase membrane stability in dystrophin-deficient myotubes, which is probably the main reason why *mdx/Sirt6^mKO* compound mutant mice show a profound improvement of muscle function. Further support for this hypothesis comes from the generation of *Utrn/mdx/Sirt6^mKO* triple mutant mice, in which the beneficial effects of *Sirt6* inactivation in *mdx* mice was gone, clearly indicating that

upregulation of *Utrn* is indispensable for the improvement of muscle function after repression of SIRT6 activity.

Previous research revealed that overexpression of utrophin in *mdx* mice substantial improves the dystrophic phenotype, thereby identifying *Utrn* as an attractive therapeutic target to treat DMD[32,46]. Expression of the *Utrn* gene is already upregulated in *mdx* mice and DMD patients as part of the repair process[14], but the level of expression is moderate, not allowing to fully compensate for the absence of dystrophin. Thus, intense efforts were made to stimulate *Utrn* expression via activation of different signaling pathways, e.g., using small molecules such as SMT C1100, which increases expression of *Utrn* about two-fold in cell culture and in *mdx* mice[11]. In principle, all patients with loss of dystrophin should benefit from increased *Utrn* expression, irrespective of the type of dystrophin mutation, which is a significant advantage compared to gene therapy approaches, aiming to correct individual mutations. Our findings revealed that suppression of SIRT6 activity was sufficient to increase *Utrn* expression to a level that improved skeletal muscles function in dystrophic *mdx* muscles, suggesting that manipulation of gene expression at the epigenetic level is a realistic option for the treatment of DMD. The regulatory elements driving *Utrn* expression have been characterized extensively, leading to the identification of an essential DUE, which is located 9 kb downstream in intron two in humans[47].

Our data indicated that SIRT6 deacetylates H3K56ac residues within the putative mouse DUE, which is 84% identical to DUE motif of human *UTRN* gene and located in intron 1. Interestingly, an increase of H3K56ac at the DUE in *Sirt6* mutants was already detected in MuSC, although expression of UTRN increased only in myofibers but not in MuSCs, which is consistent with a previous study demonstrating that DUE activity is required for *Utrn* expression only in differentiated myotubes[47]. It seems plausible that epigenetic changes in MuSC precede transcriptional changes that only occur in myotubes, probably because critical transactivating factors are only available in myotubes.

In addition to increased *Utrn* expression we also observed increased levels of *Mstn* in *mdx/Sirt6^mKO^* mice. The increase in of *Mstn* expression may be a secondary event due to *Utrn*-dependent membrane stabilization, which prevents countermeasures in *mdx* mice such as increased myofiber hypertrophy. In support of this hypothesis, overexpression of *Utrn* increases serum concentrations of MSTN, concomitant with improved muscle morphology[48]. Alternatively, the *Mstn* gene might be a direct target of SIRT6 and increased *Mstn* expression may be beneficial for the course of disease in *mdx* muscles. Indeed, we observed that *Sirt6* inactivation resulted in elevated H3K56ac levels at proximal promoter/enhancer regions of the *Mstn* gene, which is in line with a previous study showing that SIRT6 negatively regulates *Mstn* expression in C2C12 cells[34]. On the other hand, inhibition of MSTN signaling, which enhances skeletal muscle mass, was proposed as potential treatment for DMD and clinical studies were launched[49]. However, such studies yielded a mixed outcome, despite some spectacular effects in mice, raising doubts about such a strategy[50]. In particular, it was argued that larger myofibers that form as a consequence of MSTN inhibition might be more vulnerable to physical stress[51]. In addition, muscle hypertrophy is only present on some genetic backgrounds in the *mdx* mouse model but not in *D2-mdx* mice and in human DMD patients[52]. Potential beneficial effects of MSTN activation or repression might therefore by strain- and species-dependent. Notwithstanding this discussion, we currently cannot exclude that increased *Utrn* and *Mstn* expression have synergistic effects for the improvement of muscle pathology in *mdx* mice. The analysis of *Sirt6^mKO^/mdx* mice also revealed reduced activation MuSCs and partial normalization of gene expression profiles. We assume

that these changes represent secondary effects caused by increased myofiber stability, in particular since we did not observe an upregulation of *Utrn* in MuSC. However, we cannot not rule out that *Sirt6* has direct beneficial effects on MuSC function.

In conclusion, we demonstrated that epigenetic manipulation of *Utrn* expression is a viable option to improve membrane stability of myofibers in DMD. We were surprised that inactivation of *Sirt6* serves such a function, since impaired expression or function of SIRT6 has been implicated in pathogenic changes of different organs, including liver, heart and brain[45,53,54]. On the other hand, recent studies suggest that inactivation of *Sirt6* improves the course of several diseases, indicating that SIRT6 acts in a highly tissue- and context-dependent manner[55,56]. Although it will be difficult to target *Sirt6* pharmacologically in a muscle-specific way, our study paves the way for epigenetic manipulation of *Utrn* expression to treat DMD.

## Methods

**Animals.** *Sirt6^flox/flox^* mice were generated in house by flanking exon 4 to 6 of the *Sirt6* gene with two loxP sequences (Supplementary Fig. 2a). C57BL/10ScSn-Dmdmdx/J (*mdx*) mice and B10ScSn.Cg-Utrntm1Ked Dmdmdx/J were obtained from The Jackson Laboratory (Bar Harbor, ME). *ROSA26-YFP* mice were obtained from Frank Constantini (Columbia University, USA, New York City). Generation of *Pax7ICN* mice and *Pax7ZsGreen* mice have been described previously[57]. Primers used for genotyping are listed in Supplementary Data 1. All mice were maintained in individually ventilated cages, at 22.5 °C ± 1 °C and a relative humidity of 50% ± 5% with controlled illumination (12 h dark/light cycle). Mice were given ad libitum access to food and water. All animal experiments were done in accordance with the Guide for the Care and Use of Laboratory Animals published by the US National Institutes of Health (NIH Publication No. 85-23, revised 1996) and approved by the Committee for Animal Rights Protection of the State of Hessen (Regierungspraesidium Darmstadt) with the project numbers B2/1125, B2/1137 and B2/2019.

**Plasmids.** PCR products containing full-length wild-type or catalytic dead (H133Y) SIRT6 from pCMV-SPORT6-SIRT6 (Open Biosystems; GE Health-care, CO, USA) and P300 core wild-type domain (from pCMVb-p300-myc, Addgene plasmid: 30489) or inactive catalytic domain (from pCMVb-p300.DY-myc, Addgene plasmid: 30490) were cloned in Fuw-dCas9-dead Tet1CD-P2A-BFP (Addgene plasmid: 108246) with AscI and EcoRI sites (for *Sirt6*) or BlpI and EcoRI sites (for *p300* core) to replace the Tet1CD sequence. gRNA expression plasmids were generated by inserting DNA fragments containing individual gRNA sequences (Supplementary Table 3) into pLV_esiCRISPR_mCherry (Eupheria Biotech) with NdeI and EcoRI sites using established protocols[58]. The PCR product containing full-length HA-SIRT6-V5- tagged was cloned into the lentiviral vector pLJM1-EGFP (Addgene plasmid: 19319) using NheI and EcoRI sites to replace EGFP.

**Muscle regeneration assay.** Mice were anaesthetized intraperitoneally with 10% ketamine and 2% xylazine diluted with 0.9% NaCl (100 µl/10 g body weight). To introduce a muscle injury, 50 µl of 0.06 mg/ml cardiotoxin from Naja mossambica mossambica (Sigma, Germany) in 0.9% NaCl was injected into the tibialis anterior of adult mice using an insulin syringe. The needle was inserted deep into the muscle longitudinally toward the knee from the ankle. The anterior tibial muscles were analyzed 14 days after injection.

**Evan's blue staining.** To quantify sarcolemma stability, mice were intraperitoneally injected with 1% EBD (Sigma, St Louis, MO, USA) (w/v) (0.01 ml per 10 g of body mass) in PBS, pH 7.5, sterilized by passage through a Millex®-GP 0.22 µm filter (Millipore, Bedford, MA, USA). After injection, animals were returned to their cage before harvesting the diaphragm, 16 h after EBD injection. Cross-sections were cut at 8 µm, and EBD was detected as red auto-fluorescence by using a Zeiss AxioObserver Z1 fluorescence microscope.

**Serum creatine kinase assay.** Mice were anaesthetized as described above and blood samples were taken from orbital venous plexus by inserting a capillary into the medial canthus of the eye (30 degree angle to the nose). After incubation for 15 min on room temperature, blood samples were centrifuged at 4000 × g for 15 min at 4 °C and the sera were collected in new 1.5 ml tube. Serum creatine kinase (CK) activity was measured by IDEXX laboratory (Im Moldengraben 65, Kornwestheim Germany).

**Metabolic phenotyping with PhenoMaster cage.** For metabolic studies, mice were housed individually in metabolic cages (PhenoMaster) and acclimatized for

48 h before readings were taken. Metabolic parameters such as respiratory exchange rate (RER) and EE, food and liquid intake, spontaneous locomotor activity were monitored for 48 h.

**Magnetic resonance imaging (MRI)**. All MRI experiments were performed on a 7.0-T superconducting magnet (Bruker Biospin, Pharmascan, 70/16, 16 cm; Ettlingen, Germany) equipped with an actively shielded imaging gradient field of 300 mT m$^{-1}$. The frequency for the 1H isotope is 300.33 MHz. A 60-mm inner diameter linear-polarized 1H volume resonator was used for RF pulse transmission and signal reception (Bruker Biospin). Localized images were acquired using a spin-echo sequence and corrections of slice angulation were performed, if necessary. Rapid acquisition with relaxation enhancement sequences (repetition time (TR) = 2500 ms, echo time (TE) = 36.7 ms, slice thickness = 1 mm) in axial and coronal orientation were used to determine exact positioning of the lower part of the mouse body. A coronal multi-slice-multi-echo-spin-echo-sequence with an echo time TE = 8.6 ms, repetition time TR = 453 ms, a field of view FOV = 7 × 7 cm$^2$, matrix size MTX = 512 × 256, and a slice thickness of 1 mm was recorded. Volumetric quantification of fat and muscle tissue from images was processed by ImageJ. A list of anatomically defined landmarks was used to derive tissue-specific signal intensity thresholds and to define the region of interest. The tissue voxel volumes inside the region of interest were determined as cubic millimeters for each tissue class. Mice were measured under volatile isoflurane (1.5–2.0% in oxygen and air with a flow rate of 1.0 l min$^{-1}$) anesthesia; the body temperature was maintained at 37 °C by a thermostatically regulated water flow system during the entire imaging protocol[59].

**Glucose tolerance test (GTT)**. To determine glucose tolerance, mice were fasted 16 h before i.p. injection of 100 µl 20% glucose solution per 10 g body weight. Baseline blood glucose concentrations were taken before administration of glucose. Measurements after glucose injections were done at 30 min intervals over a span of 2 h with an ACCU-CHEK Active Blood Glucose System (Roche) using tail tip blood samples.

**Gene expression and western blot analysis**. Total RNA from either whole-muscle or FACS-isolated MuSCs was extracted using TRIzol reagent (Invitrogen) following instructions of the manufacturer. RNA was reverse-transcribed with PrimeScript RT Reagent Kit (TaKaRa) following standard procedures. Real-time PCR was performed using Taqman (Applied Biosystems) and Blue S'Green qPCR Kit (Biozym Blue). Relative quantitation of mRNA gene expression was performed using the ΔCT method. The Ct-values of target genes were normalized to the m36b4 housekeeping gene using the equation $\Delta Ct = Ct_{reference} - Ct_{target}$ and expressed as ΔCt. Primers are listed in Supplementary Data 1. For western blot assays, extracts from muscles, freshly isolated or proliferating MuSCs were resolved by SDS-PAGE, transferred onto nitrocellulose filters or PVDF membranes and probed with antibodies using 1:1000 dilution as indicated (Supplementary Table 4). Protein expression was visualized using a chemiluminescence detection system (GE Healthcare, Little Chalfont, United Kingdom) and quantified with the ChemiDoc gel documentation system (Bio-Rad).

**Immunofluorescence and histological analysis**. Tibialis anterior and diaphragm muscles were dissected and placed with one end in 10% gum tragacanth (Sigma-Aldrich) on a flat piece of cork before freezing in isopentane/liquid nitrogen. Frozen muscle sections (8–10 µm) were fixed in cold acetone or 4% paraformaldehyde. Immunofluorescence and H&E staining were done following standard protocols. Masson's Trichrome staining was carried out using the ACCUSTAIN® trichrome staining kit (Sigma-Aldrich). Primary antibodies for immunofluorescence staining using 1:1000 antibody dilution are listed in Supplementary Table 4. MuSCs were identified by staining for Pax7. For quantification, the numbers of Pax7-positive MuSC were counted on five section slides per animal, scoring four different areas per section.

**Isolation, cultivation and in vitro differentiation of MuSC**. Satellite cell isolation and purification were performed according to established methods[60] with minor modifications. Briefly, limb and trunk muscles were minced, digested with 100 CU Dispase (BD) and 0.2% type II collagenase (Worthington Biochemicals), and consecutively filtered through 100, 70, and 40 µm cell strainers (BD). Cells were collected by centrifugation at 1200 × g for 7 min. Pellets were re-suspended in 1.5 ml red blood cell lysis buffer containing 5 µg/ml DNase I and incubated on ice for 3 min. Subsequently, the cell suspension was filled up to 7 ml with DMEM medium containing 2% FCS, before cells were spun down. To enrich for MuSCs, isolated cells were incubated with APC fluorescence coupled primary antibodies against Sca1, CD45, CD31 (1:100 dilution in FACS-sorting buffer) for 40 min at 4 °C. After addition of 5 ml of DMEM medium containing 2% FCS, cells were spun down and the cell pellets were resuspended in 200 µl FACS-sorting buffer, before incubation with 30 µl of anti-APC micro beads (MACS) for 15 min on 4 °C. Microbeads containing Sca1+/CD45$^+$/CD31$^+$ cells were isolated by a 25 LS separation column using the QuadroMACS separator (Miltenyi Biotec). Sca1$^-$/CD45$^-$/CD31$^-$ cells were spun down, stained with Integrin-a7-FITC and CD34-Alexa Fluor 405 antibodies (1:100 dilution in FACS-sorting buffer). Integrin-a7$^+$/

CD34$^+$, GFP$^+$ (from Pax7/zsGreen mice) or YFP$^+$ (from Pax7ICNCre/ROSA-YFP mice) satellite cells were isolated using a FACS AriaIII (BD Biosciences). Gating strategies are shown in Supplementary Figs. 9 and 10, respectively. Antibodies used for FACS-sorting are listed in Supplementary Table 4. FACS-sorted MuSCs were cultured in DMEM medium with 20% FCS and bFGF (5 ng/ml). MuSCs were induced to differentiate at 95–100% confluence, 48 h after transduction using differentiation medium (DMEM medium containing 2% horse serum).

**Cultivation and in vitro differentiation of human myoblasts**. The initial biopsies, from which the cell lines were generated, were provided by MyoBank, the tissue bank of the Institut de Myologie in Paris, affiliated with EuroBioBank. MyoBank has received approval from the French Ministry of Higher Education, Research and Innovation to distribute human samples for research (Authorization code AC-2019-3502). Human myoblasts were cultured in a growth medium consisting of 199 medium and DMEM (Invitrogen Carlsbad, CA) in a 1:4 ratio, supplemented with 20% FCS (Invitrogen), 2.5 ng/ml hepatocyte growth factor (Invitrogen), 0.1 µmol/l dexamethasone (Sigma-Aldrich, St. Louis, MO, USA) and 50 µg/ml gentamycin (Invitrogen). Human myoblasts were differentiated by replacing growth medium with DMEM supplemented with 100 µg/ml transferrin, 10 µg/ml insulin and 50 µg/ml of gentamycin[61]. Origins and characterizations of human myoblast cell lines are provided in Supplementary Table 5.

**Cultivation and in vitro differentiation of C2C12 myoblasts**. Mouse C2C12 myoblasts were obtained from the ATCC (#CRL-1772) and were cultured in 4.5 g glucose containing DMEM supplemented with 10% FCS and 1% penicillin/streptomycin at 37 °C and 5% CO$_2$ at a cell concentration between 1.5 × 10$^5$ and 1.0 × 10$^6$ viable cells/75 cm$^2$. The cells were passaged by trypsinization (0.5% trypsin in 0.5 mM EDTA, Gibco BRL) from the culture plate at 70% confluence. C2C12 cells were differentiated using differentiation medium contained 2% horse serum instead of 10% FCS for 3 days.

**Lentiviral transduction of MuSC and ESC**. FACS-sorted MuSCs was cultured in DMEM medium with 20% FCS and bFGF (5 ng/ml). Before seeding of MuSCs, plates or dishes were coated with matrigel at 37 °C for 1 h and then air-dried. In total, 3000 cells/well were seeded in 94-well plate, 2 × 10$^4$ cells/well in 12-well plate and 15.0 × 10$^4$ cells in one 6 cm dish for CK assays, RT-qPCR and ChIP-qPCR experiments, respectively. Upon reaching 50% confluence cells were co-transduced with Sirt6/p300–dCas9 and non-targeting or gRNAs targeting DUE lentiviruses (Supplementary Table 3). MuSCs were induced to differentiation 48 h after transduction using differentiation medium (DMEM medium containing 2% horse serum). Cultured wildtype MuSCs were transduced with HA-SIRT6-V5 lentivirus in 10 cm dishes. Forty-eight hours after transduction, differentiation of MuSCs was induced by culture in differentiation medium (DMEM medium containing 2% horse serum). Lentivirus mediated Sirt6 knockdown in mESC (V6.5) was performed with shRNA (NM_181586.2-444s1c1: CCGGTCCCAAGTGTAA-GACGCAGTACTCGAGTACTGCGTC TTACACTTGGGATTTTTG), following standard protocols (The RNAi Consortium Broad Institute, The RNAi Consortium). Briefly, mESCs were harvested and the cell pellet was re-suspended in lentivirus-containing medium with 8 µg/ml polybrene, before plating on a monolayer of feeder cells. After 24 h incubation, the medium was replaced with growth media containing 2 µg/ml puromycin for another 4 days before further analysis.

**Hypo-osmotic stress CK release assay**. The experiment was performed following established protocols[62]. Briefly, differentiated myotubes were incubated with 132 mosmol (~50 mM sucrose) hypo-osomolar solution for 20 min at 37 °C. CK activity in the supernatant or cell lysate was measured in triplicate or quadruplicate using the Creatine Kinase-SL kit (Sekisui Diagnostics) according to the manufacturer's instructions.

**Electron microscopy (EM) and heterochromatin quantification**. Ultrastructure of MuSC in skeletal muscle was analyzed by EM as described before[60]. Briefly, skeletal muscles were fixed in 3% glutaraldehyde for 12 h at 4 °C and embedded in Epon. Ultrathin sections were contrasted with uranyl acetate and lead citrate, and analyzed using a Philips CM10 electron microscope. The content of heterochromatin was determined using the ImageJ program relative to the total cross-sectional area of MuSC nuclei.

**Chromatin immunoprecipitation (ChIP) in muscle**. ChIP assays were performed according to published protocols with some modifications[59]. Briefly, muscle tissue (gastrocnemius muscle) was chopped, washed with PBS and spun down at 1700 g for 5 min. Pellets were re-suspended in 500 µl buffer containing 0.32 M Sucrose, 3 mM CaCl, 2 mM magnesium acetate, 0.1 mM EDTA, 1 mM DTT, 10 mg/ml BSA, 10 mM Tris-HCl, pH 8 with protease inhibitors and homogenized for 5 min with metal beads and afterwards with a 7 ml dounce homogenizer. Released myonuclei were transferred into 1.5 ml Eppendorf tubes and spun down at 1700 g for 5 min. Myonuclei were cross-linked with 1% formaldehyde in PBS for 10 min, and then quenched by addition of 0.125 M glycine for another 10 min. Cross-linked myonuclei were washed with ice-cold PBS for two times and spun down at 5000 g for

5 min. Pellets were snap-frozen in liquid nitrogen, and stored at −80 °C for further use or directly employed for chromatin IP.

**ChIP-seq and data analysis of mESCs**. Undifferentiated mESCs infected with lentivirus expressing scramble or Sirt6 shRNAs were cross-linked with 1% formaldehyde/PBS for 10 min, and then were quenched by addition of 0.125 M glycine. After washing with ice-cold PBS for three times, cells were spun down and resuspend in ice-cold Cell Lysis Buffer (5 mM HEPES pH8, 85 mM KCl, 0.5% NP-40, Protease inhibitors). Following incubation on ice for 10 min, cells were pelleted, re-suspended in 150 µl Nuclear Lysis Buffer (50 mM Tris-HCl, pH 8.0, 10 mM EDTA, 1% SDS, protease inhibitors, freshly prepared) and incubated for 10 min on ice. Genomic DNA was sonicated with a Bioruptor (Diagenode) to generate 0.3–0.5 kb fragments. Sheared chromatin was diluted 10-fold with IP buffer (20 mM Tris-HCl, pH 8.0, 150 mM NaCl, 2 mM EDTA, 1% Triton X-100, complete protease inhibitor cocktail) to a final concentration of 30 µg/ml DNA. Chromatin lysates were precleared with protein A agarose beads at 4 °C for 2 h and the supernatant was incubated at 4 °C overnight with H3K56ac antibodies (Epi-Gentek). The DNA-histone-antibody complexes were precipitated by incubation with protein A agarose beads pre-blocked with 5% BSA for 2 h. Beads were pelleted and successively washed twice with Low Salt buffer (20 mM Tris-HCl, pH 8.0, 150 mM NaCl, 2 mM EDTA, 1% Triton X-100, 0.1% SDS), High Salt buffer (20 mM Tris-HCl, pH 8.0, 500 mM NaCl, 2 mM EDTA, 1% Triton X-100, 0.1% SDS), LiCl2 buffer (10 mM Tris-HCl, pH 8.0, 250 mM LiCl, 1% Na-deoxycholate, 1% NP40) and TE buffer (10 mM Tris-HCl pH 8.0, 1 mM EDTA). DNA was eluted with 500 ul freshly prepared IP Elution Buffer (100 mM NaHCO3, 1% SDS) for 30 min in room temperature on a rotating wheel. Reverse crosslinking was performed in 0.3 M NaCl while shaking at 65° for 6 h, followed by treatment with 20 ug/ml RNase (DNase free) at 37 °C for 1 h, and with 0.01 M EDTA, 0.04 M Tris pH6.8 and Proteinase K, at 45° for 1 h. ChIPed DNA was purified using mini-elute PCR kits (Qiagen) and quantified by the Qubit dsDNA HS Assay Kit (Thermo Fisher Scientific). In total, 0.5–10 ng of DNA was used as input for the TruSeq ChIP Library Preparation Kit (Illumina) with the following modifications. Instead of gel-based size selection before the final PCR step, libraries were size selected using the SPRI-bead based approach after a final PCR with 18 cycles. Samples were first cleaned with a 1× bead:DNA ratio to eliminate residuals from PCR reaction, followed by a two-sided-bead cleanup step with an initial 0.6× bead:DNA ratio to exclude larger fragments. Supernatant was transferred to a new tube and incubated with additional beads with a 0.2× bead:DNA ratio to eliminate smaller fragments. Bound DNA samples were washed with 80% ethanol, dried and resuspended in TE buffer. Library integrity was verified with the LabChip Gx Touch 24 (Perkin Elmer). Sequencing was performed with the NextSeq500 instrument (Illumina) using v2 chemistry with 1 × 75bp single end setup.

FASTQ files were controlled for quality issues, using FastQC. Quality aware trimming of reads and adapter removal were performed using Trim Galore. Read alignment against the mm10 mouse genome was done against a reference downloaded as a pre-compiled BWT index from Illumina's iGenome repository. Read alignment was performed by using bowtie version 1.1.2 with parameters -k 1 -m 1. Duplicate removal was performed by using Picard's MarkDuplicates function. Coverage vectors were generated with Deeptools bamCoverage function, using RPKM (reads per kilo base per million mapped reads) normalization[63]. Visualization of binding profiles was done by using the R/BioConductor package Gviz or the Integrative Genome Viewer[64]. Peak calling was done using MACS2[65]. The resulting set was filtered against blacklisted chromatin regions, as detected by ENCODE. Read counts across peaks were determined using the featureCounts function of the Subread package. Differential binding analysis was performed by using DESeq2 after merging overlapping peaks into reference peak sets, using the reduce function of the BioConductor GenomicRanges package[66]. H3K56ac data were compared to available data for histone modifications and other chromatin modifiers in mES cells from ENCODE[67] by downloading the raw sequencing reads of the first replicate for H3K27ac, H3k4me1, H3K4me3, H3K9ac, H3K36me3 and p300 through the UCSC browser data download portal (https://hgdownload-test.gi.ucsc.edu/goldenPath/mm9/encodeDCC/wgEncodeLicrTfbs). Reads were processed as described above. Read counts across H3K56ac peaks and H3K56ac peaks significantly increased in Sirt6^mKO were determined using featureCounts and normalized by scale normalization within R. Wilcoxon rank sum tests were performed within R using the wilcox.test function with the alternatives "greater" or "less".

**ChIP-seq and data analysis in MuSCs**. FACS-sorted MuSCs were cross-linked with 1% formaldehyde/PBS for 10 min, before quenching with 0.125 M glycine. Cross-linked cells were washed with ice-cold PBS for three times and spun down. Chromatin preparation and protein-DNA complex immunoprecipitation was carried out according to established protocols[68]. ChIPed DNA was purified using mini-elute PCR kits (Qiagen). DNA concentration was measured using Qubit® Fluorometric Quantitation (Life Science). ChIP-seq libraries were generated using the Ion ChIP-Seq Library Preparation Kit (Thermo Fisher) following standard protocols. Libraries were analyzed with the Bioanalyzer 2100 and sequenced using the IonTorrent Proton platform with Ion PI Sequencing 200 Kit v2 (Thermo Fisher). Resulting raw reads were assessed for quality, adapter content and duplication rates with FastQC. Data processing of sequencing reads in FASTQ

format were performed as described above for mES cells. All downstream analyses were carried out with R/BioConductor (http://www.bioconductor.org) using the packages GenomicRanges and GenomicFeatures. In order to compare H3K56ac to other histone modifications public data sets for H3K4me3 and H3K27ac in MuSCs[69] were downloaded from GEO (time-point T3 from GSE103163: GSM2756408, GSM2756400 and GSM2756402).

**ATAC-seq**. In total, 50,000 cells were used for ATAC Library preparation using Tn5 Transposase from Nextera DNA Sample Preparation Kit (Illumina). Cell pellet was resuspended in 50 µl PBS and mixed with 25 µl TD-Buffer, 2.5 µl Tn5, 0.5 µl 10% NP-40 and 22 µl water. Cell/Tn5 mixture was incubated at 37 °C for 30 min with occasional snap mixing. Transposase treatment was followed by 30 min incubation at 50 °C together with 500 mM EDTA pH8.0 for optimal recovery of digested DNA fragments. For neutralization of EDTA 100 µl of 50 mM MgCl2 was added followed by purification of the DNA fragments by MinElute PCR Purification Kit (Qiagen). Amplification of Library together with Indexing was performed as described elsewhere[70]. Libraries were mixed in equimolar ratios and sequenced on NextSeq500 platform using V2 chemistry with paired-end mode. Quality control of FASTQ files was done using FastQC. Quality aware trimming of reads and adapter removal were performed using Trim Galore. Reads were aligned to a pre-compiled index of the mouse mm10 genome using BWA with default settings. Duplicate reads were removed with Samtools' rmdup function. Virtual footprinting for motif analysis was done by adapting the HINT subroutine of the Regulatory Genomics Toolbox suite (http://www.regulatory-genomics.org/). Peaks were called by using MACS2 (version 2.1.1.20160309) with the default setting of the p value threshold to -p 0.01 and using the -bampe option. Coverage bigwig files were generated with Deeptools bamCoverage function at 50 bp resolution.

**DNA methylation-sensitive PCR**. TA muscle tissue was digested over night at 50 °C with digestion buffer (100 mM NaCl, 10 mM Tris-HCl, pH 8, 25 mM EDTA pH 8, 0.5% Sodium dodecyl sulfate, and freshly added 0.1 mg/ml proteinase K). After tissue digestion, phenol/chloroform/isoamyl extraction of DNA was performed. In total, 1 µg purified DNA was subjected to enzymatic digestion with methylation-sensitive enzymes HpaII and MspI (NEB) followed by qPCR with primers flanking the restriction site (Supplementary Data 1). DNA input was measured by amplification of an unrelated fragment that contains no restriction sites.

**RNA-seq and data analysis in MuSCs and muscle tissue**. Total RNA was isolated from muscle tissue or FACS-sorted MuSCs by the miRNeasy micro kit (Qiagen) together with on-column DNase digestion (DNase-Free DNase Set, Qiagen) and followed by QC on Bioanalyzer 2100 (Agilent). For MuSCs, Total RNA libraries were obtained by rRNA depletion with RiboMinus Eukaryote System v2 (Thermo Fisher) followed by Ion Total RNA-Seq Kit v2 (Thermo Fisher) according to the manufacturer's protocol. Library quality was controlled by Bioanalyzer 2100 and final sequencing was performed with Ion PI Sequencing 200 Kit v2 (Thermo Fisher). For muscle tissues, integrity of RNA and library preparations were verified with LabChip Gx Touch 24 (Perkin Elmer). In total, 300 ng of total RNA was used as input for VAHTS Stranded mRNA-seq. Library preparation following manufacture's protocol (Vazyme). Sequencing was performed on NextSeq500 instrument (Illumina) using v2 chemistry, resulting in average of 30 M reads per library with 1 × 75bp single end setup.

Raw reads were assessed for quality, adapter content and duplication rates with FastQC 0.10.1, trimmed by Reaper version 13–100[71] and terminally aligned to the Ensemble mouse genome version mm10 (GRCm38) by STAR 2.4.0a[72]. The number of reads aligning to genes was counted with featureCounts 1.4.5-p1 tool from the Subread package[73]. Only reads mapping at least partially inside exons were admitted and aggregated per gene. Reads overlapping multiple genes or aligning to multiple regions were excluded. DEGs were identified using DESeq2 version 1.62 l[74]. For MuSCs, only genes with a minimum fold change of ±2, a maximum Benjamini-Hochberg corrected p value of 0.05, and a minimum combined mean of 5 reads were assumed to be significantly differentially expressed. For muscle tissues, only genes with a maximum uncorrected p value of 0.05, and a minimum combined mean of 5 reads were assumed to be significantly differentially expressed. The Ensemble annotation was enriched with UniProt data (release 06.06.2014) based on Ensembl gene identifiers (Activities at the Universal Protein Resource (UniProt)). Correlations of replicate gene counts were assessed with the Spearman ranked correlation algorithm included in R 3.11 (R: A language and environment for statistical computing). Volcano plots were computed using the script run_DE_analysis.pl included in Trinity version 20140717, which employs R functions for plotting[75]. Further global clustering of samples was performed using the regularized-logarithm transformation method of DESeq2, based on complete euclidean distances and hierarchical clustering. Gene ontology analysis was performed using DAVID Bioinformatics Resources (http://david.abcc.ncifcrf.gov) and Enrichr (https://maayanlab.cloud/Enrichr/)[76,77]. Heat maps were generated using function "heatmap.2" in R package "gplots", using the version R-3.6.1.

**Statistical analysis**. For all quantitative analyses, a minimum of three biological replicates were analyzed with the exception of only two biological replicates for

ChIP-seq, RNA-seq and ATAC-seq data. Statistical tests were used based on the assumption that sample data are derived from a population following a probability distribution based on a fixed set of parameters. T-tests were used to determine statistical significance of differences between two groups. One-way ANOVA were used to perform multiple comparisons. The following values were considered to be statistically significant: $*p < 0.05$, $**p < 0.01$, $***p < 0.001$, $****p < 0.0001$. Calculations were done using the GraphPad Prism 8 software and R. Error bars always represent standard error of the mean (mean ± SEM). No statistical method was used to predetermine sample size.

**Reporting summary**. Further information on research design is available in the Nature Research Reporting Summary linked to this article.

## Data availability

Raw and processed ChIP-seq data in mESCs and MuSCs are available in the NCBI Gene Expression Omnibus (GEO), under accession number GSE168331 and GSE168330, respectively. ATAC-seq data were deposited under accession number GSE168329. RNA-seq data of MuSCs and muscle tissue were deposited under accession number GSE199487, GSE168983 and GSE168984, respectively. Published data were downloaded from GEO time-point T3 from GSE103163: GSM2756408, GSM2756400 and GSM2756402. All other data generated during the current study are available from the corresponding author on reasonable request. Source data are provided with this paper.

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

## Acknowledgements

We thank Sonja Krüger, Kerstin Richter and Barbara Zimmermann for technical help as wells as Astrid Wietelmann and Ursula Hofmann for MRI analysis. We are grateful to Shuichi Watanabe and Krishna-Moorthy Sreenivasan for the help with immuno-fluorescence staining and inspiring discussions, respectively. We are grateful to Vincent Mouly, Anne Bigot, and the MyoLine platform for immortalization of human cells of the Institut de Myologie (Paris, France) for supplying the human cell lines used in this study. This work was supported by the German Research Foundation (DFG) Transregional Collaborative Research Centre 81 (TP A02) (T.B. and X.Y.), the collaborative research center SFB 1213 (TP B02) (T.B. and X.Y.), the Transregional Collaborative Research Centre 267 (TP A05) (T.B.), and the Excellence Cluster Cardiopulmonary Institute (CPI) (T.B.).

## Author contributions

X.Y., Y.Z. and T.B. conceived and designed this project. A.M.G. performed most of the experiments, analyzed the data, and prepared figures. X.G. performed western blot and immunofluorescence staining. S.G. performed RNA-seq and ChIP-seq. S.G. and C.K. analyzed RNA-seq. MB performed ChIP-seq and ATAC-seq data analysis. U.G. performed electron microscopy imaging. AA conducted FACS analysis. E.B. and C.S. generated the *Sirt6* floxed mouse line. Y.Z. contributed to data analysis, discussion and advice. K.M. generated the immortalized human myoblast cell lines. A.M.G., X.Y. and T.B. wrote the manuscript.

## Funding

## Competing interests

The authors declare no competing interests.
