## [Peer Review File · Nature Communications]

Title: Inactivation of Sirt6 ameliorates muscular dystrophy in mdx mice by releasing suppression of utrophin expressionREVIEWER COMMENTS

Reviewer #1 (Remarks to the Author):

The manuscript of “Inactivation of Sirt6 ameliorates muscular dystrophy in mdx mice by releasing suppression of utrophin expression” by Georgieva et al reports that SIRT6 is important for regulation of utrophin expression, possibly as a deacetylase of H3K56ac at the DUE enhancer. In addition, the team demonstrated ameliorative effects of inactivation of Sirt6 in the context of dystrophin-deficient mdx mice. The observations are interesting, some experiments are well controlled, but additional experimentation and critical analyses are required to improve the rigor of the manuscript.

Specific comments:

1. Some wording needs to be revised to avoid overstating. While data shows beneficial effects of inactivation of SIRT6 on mdx mice, words of “normalization, normalized, reverts, restored, returned etc. are used through the text, which is overstating and can be miss leading.
2. Western blots in Fig. 1D, F and G are over exposed, judging by band images, loading control might not from the same stripped membrane, or orientation of the image flipped (Fig. 1D, G, QSC), and quantification of repeats (n = 3) is required.
3. Fig 1E, DMD patients n = 3, would be nice to have n = 3 for Fig. 1F and quantified as well.
4. Fig. 3C, D, why data on control mice not shown.
5. Fig. 4F, Dab2 and Ptpn22 expression presented as significantly different between control and Sirt6mKO/mdx, how to justify for the statement of “reverted to control level, suggesting an overall normalization of cellular physiology in Sirt6mKO/mdx mice”. Also, what is the statistical analysis on Hspa2 expression between control and Sirt6mKO/mdx, not significant?
6. Bioinformatic analysis shown is quite superficial. Analysis of the distance of H3K56ac peaks in relation to nearest TSS will help to illustrate genome-wide distribution of H3K56ac marks. Statistical analyses of boxplot need to be included (Fig. 5B, C, Fig. S4B, C, D).
7. qChIP experiments need to be quantified as percentage of enrichment relative to input chromatin DNA (Fig. 5I, 6C, F, Fig. S5D, Fig. S6D, E).
8. In terms of regulatory mechanism, qChIP analyses to show the association of AP1, p300 and SIRT6 to the DUE enhancer are needed to provide insight.

Reviewer #2 (Remarks to the Author):

The important finding reported in the paper is that SIRT6 deacetylates H3K56ac in myofibers to suppress expression of utrophin, the dystrophin-related protein known to stabilize the sarcolemma in absence of dystrophin and attenuate the pathology in the mdx mouse model of DMD. This study shows that the inactivation of Sirt6 in dystrophin-deficient mdx mice reduced damage of myofibers, reduced dystrophic muscle pathology, and improved muscle function, leading to attenuated activation of muscle stem cells (MuSCs). ChIP-seq analysis and CRISPR targeting of the target locus for binding of SIRT6 showed that

SIRT6 is critical for removal of H3K56ac at the downstream utrophin Enhancer (DUE), which is indispensable for utrophin expression.

The authors focussed on Sirt6 following the analysis of the expression of different sirtuins in muscles of mdx mice which showed decreased Sirt1 but increased Sirt6 expression both in quiescent MuSC and in muscle fibers. Pax7-Cre mediated inactivation of Sirt6 in mdx mice normalized pathological features of dystrophic muscles and attenuated persistent activation of MuSC, whereas no obvious morphological and functional abnormalities were detected in skeletal muscles of Sirt6 mutants. Critically, the absence of Sirt6 resulted in massive hyper-acetylation of H3K56, indicating that SIRT6 is the dominant H3K56ac deacetylase in skeletal muscle. They went on to demonstrate that SIRT6 suppresses Utrn expression in dystrophic muscles via deacetylation of H3K56ac at the downstream utrophin enhancer (DUE). To definitely prove that upregulation of utrophin is responsible for attenuation of the dystrophic pathology in Sirt6mKO/mdx, the authors inactivated the Utrn gene in Sirt6mKO /mdx mutant mice, yielding Sirt6mKO/Utrn-/- /mdx mutants which abolished the effect. improved.

The suppression of SIRT6 activity was sufficient to increase Utrn expression to a level that improved skeletal muscles function in dystrophic mdx muscles, suggesting that manipulation of gene expression at the epigenetic level is a realistic option for the treatment of DMD. The authors admit that it will be difficult to target Sirt6 pharmacologically in a muscle-specific way, but this study suggests that epigenetic manipulation of Utrn expression may be a possibility for the therapy of DMD.

The authors only present mRNA levels data for utrophin. In order to compare these data with studies of increased utrophin in the literature and to assess its potential for treatment of DMD, it would be useful to see what this corresponds to at the utrophin protein level.

There is a typo in Figure 5 B labelling. Mous should be mouse.

.

Reviewer #3 (Remarks to the Author):

Georgieva and colleagues identify Sirt6 as an upregulated gene in muscle cells of mdx mice. From that point, the authors present a number of experiments suggesting that Sirt6 downregulation in skeletal muscle is enough to alleviate muscular dystrophy in mice. These effects might in part be achieved due to the influence of Sirt6 in the regulation of the utrophin gene (Utrn). In this sense, the authors demonstrate that Sirt6 represses H3K56 acetylation in the enhancer region of the utrophin gene. Thus, Sirt6 deletion leads to increased utrophin expression, which compensates for the lack of dystrophin in the mdx mice.

The manuscript is very well written and constitutes an experimental tour de force using multiple genetically engineered mouse models and transcriptomic analyses. Some aspects, however, remain unconvincing and require some consolidation. My major points can be found below:

1/ SIRT6 whole-body KO mice are characterised by a massive glycolytic rewiring. While the authors show no difference between control and SIRT6mKO mice on an GTT test, this does not rule out a high reliance on glycolysis, which could influence MuSC metabolism and their ability to activate in disease settings. Therefore, it would be important to characterise, at least at the gene expression level, if glycolytic markers are altered. In addition, it would be interesting to explore Hif-1 recruitment to target genes, which is generally impeded by Sirt6.

2/ The way the transcriptomic data and analyses are presented is a bit vague and uninformative. Their main conclusion is the identification of some global paths influenced by the different mutations and the quantification of how many genes equally behave or not in the different groups tested. For the most prominent differences, some physiological implication of the findings would be required. For example, in Fig.1B gene sets as "apoptotic process", "osteoblast differentiation" and "response to cAMP" appear, yet no physiological implication in the context of mdx is clearly demonstrated. Further, these gene sets include often positive and negative regulators. Therefore, some validation for a physiological change in these functions would be needed, at least for the top scorers. Similar comments could be applied to the analyses in Fig.2 and Fig.4.

3/ How are the histone acetylation profiles for the experiment in Fig. 2? It would be important to examine some specific markers, such as those examined in Fig.1G in order to understand the influence of the mdx mutation and SIRT6 in the overall pattern.

4/ For Fig.2C, one would expect a larger range of analyses. How many genes are unregulated in the SIRT6 mKO/mdx mice but not on the mdx? Also, how many genes are down regulated in both genotypes and what is the overlap?

5/ In Figure 3, the control group should be shown through the figure.

6/ The authors identify a number of genes controlled by SIRT6 by ChIP-seq data obtained through H3K56ac, H3K27ac or H3K4me3 pulldowns. However, the reader remains clueless on whether the regulation exerted by SIRT6 is direct or not. At least for the most critical aspect of the manuscript (the regulation of the Utrn gene), the authors should examine whether SIRT6 sits on its enhancer region, whether the binding is regulated and whether this correlates with H3K56ac levels.

7/ Figure 6 is a really elegant piece of work. However, a few groups should be added on the analyses. First, Fig.6C and 6F should display the gRNA + catalytically inactive groups. Second, similar assays should be done in MuSCs from SIRT6 mKO mice in the absence of the mdx mutation.

8/ Would CK levels increase if the experiments in Fig. 6E and 6H were performed in the absence of the mdx mutation?

9/ The experiment in Figure 7 is also very interesting. However, these mice are very poorly characterised. What is their body size, fat mass, muscle mass, muscle force, etc.? It might simply be that

the mice are in a very poor shape when they lack both the mdx and the Utrn genes. There were multiple gene expression changes that were affected by the SIRT6 mKO irrespectively of the mdx mutation. Does the Utrn mutation affects them?

Minor:

- SIRT7 mRNA levels in MuSCs from mdx mice are highly unregulated. Is this consistent at the protein level?
- In the muscle regeneration experiment shown in Supplemental Figure 1, a more detailed time course experiment would be needed, as the 14 day timing might not reflect a slower (even if effective) regeneration.
- Line 144. Where do the 1285 genes come from? The graph depicts 977.
- Lines 255-256. See point 6. The fact that H3K56 acetylation is upregulated at the DUE of the Utrn gene in SIRT6 mKO/mdx mice does not proof that SIRT6 is the enzyme directly performing this deacetylation reaction.
- What is the expression of the Utrn gene in SIRT6 mKO mice compared to control, in the absence of the mdx mutation?

REVIEWER COMMENTS

Reviewer #1 (Remarks to the Author):

The manuscript of “Inactivation of Sirt6 ameliorates muscular dystrophy in mdx mice by releasing suppression of utrophin expression” by Georgieva et al reports that SIRT6 is important for regulation of utrophin expression, possibly as a deacetylase of H3K56ac at the DUE enhancer. In addition, the team demonstrated ameliorative effects of inactivation of Sirt6 in the context of dystrophin-deficient mdx mice. The observations are interesting, some experiments are well controlled, but additional experimentation and critical analyses are required to improve the rigor of the manuscript.

Response: We thank the reviewer for the positive evaluation of the manuscript and for the helpful comments.

Specific comments:

1. Some wording needs to be revised to avoid overstating. While data shows beneficial effects of inactivation of SIRT6 on mdx mice, words of “normalization, normalized, reverts, restored, returned etc. are used through the text, which is overstating and can be miss leading.

Response: We agree with the reviewer’s comment that overstating need to be avoided. Inactivation of *Sirt6* in mdx mice has clear beneficial effects but does not completely normalize the mdx phenotype. We have toned down our statements and now use much more modest wording.

2. Western blots in Fig. 1D, F and G are over exposed, judging by band images, loading control might not from the same stripped membrane, or orientation of the image flipped (Fig. 1D, G, QSC), and quantification of repeats (n = 3) is required.

Response: The reviewer is right that of the western blots were overexposed, since endogenous Sirt6 is expressed at low levels in MuSCs and thus is difficult to detect. We have repeated several western blots, using higher protein concentration in the loaded sample and using more concentrated antibodies (Sirt6 WB in muscle in (revised Fig. 1d); H3K56ac, H3K9ac and H3 WBs in QSCs/freshly isolated MuSC (fiMuSCs) in (revised Fig. 1g)). Images of overexposed WB were replaced by new images of the same blot, exposed for a shorter time ((Fig. 1g), H3K56ac in Muscle). The GAPDH WB images of QSCs/fiMuSCs in (Fig. 1d) was from the same stripped membrane as the Sirt6 WB. The orientation of membrane was not flipped but the position of the blot was different when taking images. To avoid any confusion, we repeated the experiment and now show new results of the WB analysis in the (revised Fig. 1d) (Sirt6 & GAPDH WB in QSC/fiMuSCs).

3. Fig 1E, DMD patients n = 3, would be nice to have n = 3 for Fig. 1F and quantified as well.

Following the reviewer’s request, we have quantified all western blots (n= 3-5) using the ImageJ software. The results are shown as bar graphs the (revised Fig. 1d, g). The number of DMD patient derived cell lines were increased to n=4 and quantified. Results are shown in the (revised Fig. 1f).

4. Fig. 3C, D, why data on control mice not shown.

Response: We have revised the figure as requested. Control mice are now shown in the (revised Fig. 3c, d). We have also included a control diaphragm muscle in the (revised Fig. 3e).

5. Fig. 4F, Dab2 and Ptpn22 expression presented as significantly different between control and Sirt6mKO/mdx, how to justify for the statement of “reverted to control level, suggesting an overall normalization of cellular physiology in Sirt6mKO/mdx mice”. Also, what is the statistical analysis on Hspa2 expression between control and Sirt6mKO/mdx, not significant?

Response: The reviewer is right that Dab2 and Ptpn22 expression levels are still different between control and *Sirt6*^{mKO}/*mdx* mice. The statement “reverted to control level, suggesting an overall normalization of cellular physiology in *Sirt6*^{mKO}/*mdx* mice” referred to results shown in the heatmap (Fig. 4e), where expression of all genes in *Sirt6*^{mKO}/*mdx* mice were close to control levels. However, validation of the RNA-seq data by RT-qPCR revealed that expression of Dab2 and Ptpn22 is lower in *Sirt6*^{mKO}/*mdx* compared to *mdx* muscles but still differs significantly from wild type levels. In contrast, other genes such as Hspa2 did not show any significant difference anymore in *Sirt6*^{mKO}/*mdx* compared to control muscles. Statistical calculations indicate that differences in expression of Hspa2 is not significant between control and *Sirt6*^{mKO}/*mdx*, which is now indicated as ‘ns’ in the (revised Fig. 4f). Obviously, deletion of Sirt6 expression is sufficient to reduce expression of several genes in *mdx* muscles close to wildtype levels, while expression of other genes is only moderately affected and still differs significantly from wild type levels. We have changed the text accordingly.

6. *Bioinformatic analysis shown is quite superficial. Analysis of the distance of H3K56ac peaks in relation to nearest TSS will help to illustrate genome-wide distribution of H3K56ac marks.*

Response: Thanks to the reviewer’s comment, we made major efforts to improve the bioinformatic analysis. We analyzed genome-wide distribution of the upregulated H3K56ac peaks in *Sirt6* mutant MuSCs as well as the distance of these peaks to the nearest TSS as recommended by the reviewer. The results shown in the (revised Fig. 5c, d) clearly indicate that enrichment of H3K56ac signals in *Sirt6*^{mKO} MuSCs mainly happens within distal intergenic regions. Such distal intergenic regions often contain enhancer DNA sequences, which activate gene expression over distances of several thousand base pairs. Importantly, localization of upregulated H3K56ac peaks in *Sirt6* mutant MuSCs correlates with H3K27ac+/H3K4me3- peaks (revised Fig. 5e), indicating active enhancers.

To dissect whether the genome-wide distribution of H3K56ac peaks targeted by Sirt6 is cell-type specific, we also analyzed H3K56ac peaks that were lost upon Sirt6 knockdown embryonic stem cells (ESCs). Consistently, increased H3K56ac deposition mainly occurs at distal intergenic regions in *Sirt6* knockdown ESCs (revised Supplemental Fig. 5c), which are characterized by enhancer marks such as H3K27ac, H3K4me1 and p300 (revised Supplemental Fig. 5d). Taken together, we believe we have gathered strong evidence that Sirt6 deacetylate H3K56ac within distal intergenic enhancer regions.

Statistical analyses of boxplot need to be included (Fig. 5B, C, Fig. S4B, C, D).

Response: We apologize for this shortcoming. The statistical analyses of the boxplots were previously indicated in the figure legend and but are now incorporated in the (revised Fig. 5b, e) and (revised Supplemental Fig. 5b, d, e).

7. *qChIP experiments need to be quantified as percentage of enrichment relative to input chromatin DNA (Fig. 5I, 6C, F, Fig. S5D, Fig. S6D, E).*

Response: The reviewer is right that qChIP experiments are often quantified by calculating the percentage of enrichment relative to input chromatin DNA (%IP). However, this approach has a major disadvantage, mainly but not exclusively due to differences in handling input and ChIP samples. The input sample is taken very early from the chromatin fraction during the ChIP procedure and processed separately from the ChIP samples. Furthermore, the %IP method does not correct for differences in chromatin configuration inhibiting antibody-antigen interactions when comparing different samples. Essentially, the %IP method only corrects for technical variations and does not take into account important biological variations.

In 2007, Haring and colleague published a method for determining histone modifications on specific genes². The method normalizes IP ChIP-qPCR signals for histone modifications relative to the total histone IP signal, e.g., H3K9me3 relative to total H3 on the same DNA sequence. The method is widely used for ChIP-qPCR normalization when chromatin configurations are expected to be altered due to altered histone modifications, which will change chromatin compaction between different samples. In this study, we found a global decrease of H3 protein levels and of the linker histone H1.0 at the DUE in

Sirt6^{mKO}/mdx compared to *mdx* muscles, indicating a more open chromatin structure and lower nucleosome density, particularly at the DUE of the *Utrn* gene in *Sirt6^{mKO}/mdx* mice (Fig. 1 for reviewer). Our finding is consistent with a previous study demonstrating that H3K56 acetylation increases nucleosome unwrapping by evicting H1.0³. These results indicate that input alone is not sufficient for proper normalization.

Figure 1 for reviewer. a) WB showing the protein levels of histone H3 and GAPDH in *mdx* and *Sirt6^{mKO}/mdx* muscle. Ratios of H3/GAPDH are shown in the right panel.

b) ChIP-qPCR showing the relative enrichment of H1.0 at DUE after normalization to Histone H3 in *mdx* and *Sirt6^{mKO}/mdx* muscles (Unpaired t-test: **p < 0.01).

To properly interpret our ChIP-qPCR data shown in (Fig. 5i; 6c, f; Fig. S5d; Fig. S6d, e) (now revised Fig. 5m; Fig. 6e,h; Fig. S7d), in which H3K27ac and H3K56ac deposition on specific gene loci were evaluated, we therefore used a normalization method that factors in the nucleosome density. We hope the reviewer agrees with our line of reasoning and accepts our approach.

8. In terms of regulatory mechanism, qChIP analyses to show the association of AP1, p300 and SIRT6 to the DUE enhancer are needed to provide insight.

Response: We thank the reviewer to raise this important point. We performed ChIP-qPCR using chromatin isolated from myotubes and now demonstrate that AP1, p300, and SIRT6 specifically bind to the DUE of the *Utrn* gene. The results are shown in the (revised Fig. 5k, l).

Reviewer #2 (Remarks to the Author):

The important finding reported in the paper is that SIRT6 deacetylates H3K56ac in myofibers to suppress expression of utrophin, the dystrophin-related protein known to stabilize the sarcolemma in absence of dystrophin and attenuate the pathology in the mdx mouse model of DMD. This study shows that the inactivation of Sirt6 in dystrophin-deficient mdx mice reduced damage of myofibers, reduced dystrophic muscle pathology, and improved muscle function, leading to attenuated activation of muscle stem cells (MuSCs). ChIP-seq analysis and CRISPR targeting of the target locus for binding of SIRT6 showed that SIRT6 is critical for removal of H3K56ac at the downstream utrophin Enhancer (DUE), which is indispensable for utrophin expression. The authors focussed on Sirt6 following the analysis of the expression of different sirtuins in muscles of mdx mice which showed decreased Sirt1 but increased Sirt6 expression both in quiescent MuSC and in muscle fibers. Pax7-Cre mediated inactivation of Sirt6 in mdx mice normalized pathological features of dystrophic muscles and attenuated persistent activation of MuSC, whereas no obvious morphological and functional abnormalities were detected in skeletal muscles of Sirt6 mutants. Critically, the absence of Sirt6 resulted in massive hyper-acetylation of H3K56, indicating that SIRT6 is the dominant H3K56ac deacetylase in skeletal muscle. They went on to demonstrate that SIRT6 suppresses Utrn expression in dystrophic muscles via deacetylation of H3K56ac at the downstream utrophin enhancer (DUE). To definitely prove that upregulation of utrophin is responsible for attenuation of the dystrophic pathology in Sirt6mKO/mdx, the authors inactivated the Utrn gene in Sirt6mKO/mdx mutant mice, yielding Sirt6mKO/Utrn^{-/-}/mdx mutants which abolished the effect. improved. The suppression of SIRT6 activity was sufficient to increase Utrn expression to a level that improved skeletal muscles function in dystrophic mdx muscles, suggesting that manipulation of gene expression at the epigenetic level is a realistic option for the treatment of DMD. The authors admit that it will be difficult to target Sirt6 pharmacologically in a muscle-specific way, but this study suggests that epigenetic manipulation of Utrn expression may be a possibility for the therapy of DMD.

The authors only present mRNA levels data for utrophin. In order to compare these data with studies of increased utrophin in the literature and to assess its potential for treatment of DMD, it would be useful to see what this corresponds to at the utrophin protein level.

Response: We thank the reviewer for careful evaluation of the manuscript. We followed the advice from the reviewer and monitored protein levels of Utrophin in wildtype (control), *mdx*, and *Sirt6^{mKO}/mdx* muscles. We found increased UTRN protein levels in *mdx* muscles compared to control tissue. The increase of UTRN in *mdx* muscles is most likely caused by compensatory upregulation of the synaptic and extrasynaptic utrophin content⁴ and/or increased numbers of newly formed immature myofibers in *mdx* mice, showing transiently elevated Utrn expression⁵. Inactivation of *Sirt6* in *mdx* mice further enhances UTRN protein levels compared to *mdx* mice, which partially normalizes the *mdx* phenotype in *Sirt6^{mKO}/mdx* muscles (revised Fig. 5i).

There is a typo in Figure 5 B labelling. Mous should be mouse.

Response: We have carefully reviewed (Fig. 5b) but cannot find a typo.

Reviewer #3 (Remarks to the Author):

Georgieva and colleagues identify *Sirt6* as an upregulated gene in muscle cells of *mdx* mice. From that point, the authors present a number of experiments suggesting that *Sirt6* downregulation in skeletal muscle is enough to alleviate muscular dystrophy in mice. These effects might in part be achieved due to the influence of *Sirt6* in the regulation of the utrophin gene (*Utrn*). In this sense, the authors demonstrate that *Sirt6* represses H3K56 acetylation in the enhancer region of the utrophin gene. Thus, *Sirt6* deletion leads to increased utrophin expression, which compensates for the lack of dystrophin in the *mdx* mice.

The manuscript is very well written and constitutes an experimental tour de force using multiple genetically engineered mouse models and transcriptomic analyses. Some aspects, however, remain unconvincing and require some consolidation. My major points can be found below:

Response: We thank the reviewer for the positive evaluation of the manuscript and for the constructive comments. We have revised the manuscript following the reviewer's recommendations.

1. *SIRT6* whole-body KO mice are characterised by a massive glycolytic rewiring. While the authors show no difference between control and *SIRT6*^{mKO} mice on an GTT test, this does not rule out a high reliance on glycolysis, which could influence MuSC metabolism and their ability to activate in disease settings. Therefore, it would be important to characterise, at least at the gene expression level, if glycolytic markers are altered. In addition, it would be interesting to explore Hif-1 recruitment to target genes, which is generally impeded by *Sirt6*.

Response: We were also surprised to see in contrast to germline inactivation of *Sirt6* no differences in the GTT test in muscle-specific *Sirt6*-mutant mice. We followed the reviewer's advice and further investigated the impact of *Sirt6* inactivation on glycolytic metabolism. RNA-seq analysis of freshly isolated *Sirt6*^{mKO} MuSCs revealed no enrichment of gene ontology terms related to glycolytic metabolism in *Sirt6*^{mKO} MuSCs. We calculated the RPKMs of key glycolytic genes, which have been described as *Sirt6* target genes in embryonic stem cells⁶ and again did not find expression changes, albeit *Ldhb* was significantly downregulated (revised Supplemental Fig. 2m). In addition, we performed RT-qPCR analysis to monitor expression of several key glycolytic genes including *Glut1*, *Ldha*, and *Pdk1* in *Sirt6*^{mKO} muscle tissue. No obvious gene expression changes were detected.

Expression of *Hif1a* gene was not altered in skeletal muscles after inactivation of *Sirt6*. We have included these results in the (revised Supplemental Fig. 2n). We assume that *Sirt1* but not *Sirt6* is the key player for repressing glycolytic genes based on the following arguments: 1) Expression of *Sirt1* is much higher than *Sirt6* in quiescent MuSCs, in which FAO is dominant and glycolytic gene are repressed; 2) Expression of *Sirt1* decreases while expression of *Sirt6* remains unchanged in proliferating MuSCs, in which glycolysis becomes dominant and glycolytic genes are highly expressed⁷. Since the expression levels of potential target genes of the HIF1a/*Sirt6* axis are not altered in *Sirt6* KO MuSCs, we hope the reviewer agrees that it does not make sense to explore Hif-1 recruitment to target genes in this context, which would be a major effort.

2. The way the transcriptomic data and analyses are presented is a bit vague and uninformative. Their main conclusion is the identification of some global paths influenced by the different mutations and the quantification of how many genes equally behave or not in the different groups tested. For the most

prominent differences, some physiological implication of the findings would be required. For example, in Fig. 1B gene sets as "apoptotic process", "osteoblast differentiation" and "response to cAMP" appear, yet no physiological implication in the context of *mdx* is clearly demonstrated. Further, these gene sets include often positive and negative regulators. Therefore, some validation for a physiological change in these functions would be needed, at least for the top scorers. Similar comments could be applied to the analyses in Fig. 2 and Fig. 4.

Response: The reviewer has a point here, although it is rather uncommon to follow up all gene expression changes that are uncovered by RNAseq and subsequent GO-term analysis. To explore whether changes in gene expression were associated with physiological changes, we performed additional experiments and also investigated in more detail the consistency of observed gene expression changes and correlated the findings to functional changes. For example, to interrogate the physiological relevance of gene expression changes related to apoptosis and cell cycle activity, we performed TUNEL staining and EdU incorporation assays. We found increased numbers of TUNEL⁺ and EdU⁺ MuSCs in *mdx* mice and included these data into the (revised Supplemental Fig. 1 a, b). Since the Calcineurin-cAMP-PKA pathway contributes to the quiescence of MuSCs⁹, it is not surprising that genes involved in cAMP-signaling are deregulated in *mdx* MuSCs, which have lost quiescence due to continuous degeneration and regeneration of myofibers. We have added a sentence in the main text to point out that deregulation of Calcineurin-cAMP-PKA pathway genes corresponds to the functional changes in *mdx* MuSCs.

To gain deeper insights into differentially expressed genes (DEGs) related to 'Osteoblast differentiation', we analyzed DEGs included in this GO term (Fig. 2 for reviewer). We found that the transcription factor *Runx2*, which serves as a master regulator for osteoblast differentiation, as well as its target genes *Spp1*, *Coll1a1*, *Tnc* etc were upregulated in *mdx* MuSC. The upregulation of *Runx2* might be attributed to increased Tgfb signaling in *mdx* MuSC pathway¹², since Tgfb signaling directly activates *Runx2*. However, BMP2, its downstream target gene *Alpl* (an early bone differentiation marker) and its upstream activators *Gli1*, *Gli2*, *Cyr61*, *Tgfb3* (receptor bound by Tgfb2) were significantly downregulated. Furthermore, expression of *JunB*, which stimulates osteoblast differentiation, was also reduced. The incomplete activation of genes in *mdx* MuSCs, required for osteoblast differentiations, explains why we did not observe trans-differentiation of myoblasts into osteoblast despite a strong enrichment of the GO term "Osteoblast differentiation" in *mdx* MuSC.

Figure 2 for reviewer: Heatmap of RNA-seq data showing the deregulated genes related to 'Osteoblast differentiation' in *mdx* MuSCs.

The top hits in the GO term analysis shown in (Fig. 2a) are G1 to S cell cycle control and DNA replication. To explore whether the GO term enrichment is associated with functional changes, we performed *in vivo* EdU incorporation assay and detected increased proliferation of *mdx* MuSCs compared to control MuSCs, which was dramatically reduced in *Sirt6^{mKO}/mdx* MuSCs (revised Fig. 2d). The finding is consistent with reduced activation of MuSC in *mdx* muscle after *Sirt6* inactivation.

The upregulated genes in *mdx* mice shown in (Fig. 4c) are mainly involved in stress response signaling, including p53 signaling, oxidative damage and stress, and innate immune responses (complement activation and coagulation cascades). We demonstrate in (Fig. 3) that *Sirt6* inactivation partially normalizes these transcriptional changes but also improves corresponding physiological parameters.

We observed a downregulation of genes related to fatty acid beta oxidation in *mdx* mice (Fig. 4d). This finding is consistent with a published study demonstrating that *mdx* muscle fibers are characterized by lowered aerobic glycolysis and fatty acid oxidation¹³. Furthermore, it is well known that myofiber damage in *mdx* mice results in mitochondrial dysfunction¹⁴. Since inactivation of *Sirt6* in *mdx* reduces myofiber damage as shown in (Fig. 3e, f), it makes sense that transcriptional changes related to mitochondrial dysfunction are partially normalized in *Sirt6^{mKO}/mdx* muscles.

3. How are the histone acetylation profiles for the experiment in Fig. 2? It would be important to examine some specific markers, such as those examined in Fig. 1G in order to understand the influence of the *mdx* mutation and SIRT6 in the overall pattern.

Response: We determined histone acetylation profiles of H3K56ac, H3K9ac and H3K18ac in control, *mdx* and *Sirt6^{mKO}/mdx* MuSCs as recommended by the reviewer. We found that H3K56ac and H3K9ac levels are very low in control and *mdx* MuSCs but are significantly increased in *Sirt6^{mKO}/mdx* MuSCs. H3K18ac remains unchanged in all tested MuSCs samples. The results are shown in the (revised Supplemental Fig. 3h). The detection of increased H3K9ac levels in *Sirt6^{mKO}/mdx* MuSCs was surprising, since no change was observed in *Sirt6^{mKO}* MuSCs compared to control MuSCs. We assume that the increase of H3K9ac in *Sirt6^{mKO}/mdx* MuSCs is a secondary effect, probably due to increased HAT expression or activity. The increase of H3K9ac in *Sirt6^{mKO}/mdx* MuSC may contribute to upregulation of genes in *Sirt6^{mKO}/mdx* MuSC compared to *mdx* MuSC (revised Supplemental Fig. 3f).

4. For Fig. 2C, one would expect a larger range of analyses. How many genes are unregulated in the SIRT6 mKO/*mdx* mice but not on the *mdx*? Also, how many genes are down regulated in both genotypes and what is the overlap?

Response: We have performed additional RNA-seq experiments using *Sirt6^{mKO}* MuSCs and also re-analyzed the RNA-seq datasets. We identified 335 upregulated genes in the *Sirt6^{mKO}/mdx* MuSC that are not upregulated in *mdx* relative to control MuSC. 210 genes are downregulated in both genotypes, relative to controls. Thanks to the additional RNAseq of *Sirt6^{mKO}* MuSCs, we found that only 9 out of 335 genes, which were upregulated in *Sirt6^{mKO}/mdx* but not in *mdx* MuSC, are also upregulated in *Sirt6^{mKO}* MuSCs, indicating that most of the upregulated genes are not directly affected by Sirt6 (revised Supplemental Fig. 3 f, g.). Since RNA-seq data do not distinguish between primary and secondary events, we assume that genes, which are upregulated in *Sirt6^{mKO}/mdx* but not in *mdx* mice, are subject to secondary events that occur in *Sirt6^{mKO}/mdx* MuSC (e.g. due to enhanced myofiber stability). 210 genes are downregulated in both *mdx* and *Sirt6^{mKO}/mdx*, indicating these genes are not derepressed by Sirt6 inactivation. We have included these data in the (revised Supplemental Fig. 3i).

5. In Figure 3, the control group should be shown through the figure.

Response: We have now included control groups for all experiments shown in (revised Fig. 3).

6. The authors identify a number of genes controlled by SIRT6 by ChIP-seq data obtained through H3K56ac, H3K27ac or H3K4me3 pulldowns. However, the reader remains clueless on whether the regulation exerted by SIRT6 is direct or not. At least for the most critical aspect of the manuscript (the regulation of the *Utrn* gene), the authors should examine whether SIRT6 sits on it enhancer region, whether the binding is regulated and whether this correlates with H3K56ac levels.

Response: We agree with the reviewer that it is important to demonstrate direct binding of SIRT6 to putative targets such as the DUE of the *Utrn* gene. Unfortunately, our extensive attempts to map SIRT6-binding sites by ChIP-seq were not successful due to the low quality of commercially available antibodies. We assume that the limited material available from primary cells and the low expression of *Sirt6* in MuSCs contributed to this frustrating outcome.

To overcome these limitations, we have employed C2C12 muscle cells and also expressed HA-tagged Sirt6 in MuSCs. ChIP-qPCR experiment revealed a strong enrichment of SIRT6 at the DUE of the *Utrn* gene when using chromatin prepared from differentiated C2C12 cells. The results are shown in the (revised Fig. 5k). In addition, we performed ChIP-qPCR using chromatin from differentiated MuSC, expressing HA-Sirt6. Consistently, we detected enrichment of HA-Sirt6 at DUE of *Utrn* gene (Fig. 3 for reviewer). The results of this experiment are presented in (Fig. 5l) as well.

Figure 3 for reviewer: ChIP-qPCR analysis of SIRT6 at the DUE of the *Utrn* gene in HA-Sirt6 overexpressing MuSCs derived myotubes. (Unpaired t-test: **p<0.01, n=7).

7. Figure 6 is a really elegant piece of work. However, a few groups should be added on the analyses. First, Fig.6C and 6F should display the gRNA + catalytically inactive groups. Second, similar assays should be done in MuSCs from SIRT6 mKO mice in the absence of the *mdx* mutation.

Response: We thank the reviewer for appreciating our experimental approach. We have extended the experiments, which have included a group with gRNA + catalytically inactive enzymes (revised Fig. 6e, h) (previously Fig. 6c, f). We also performed similar epigenome editing experiments in MuSCs from *Sirt6*^{mKO} mice. We found that only recruitment of wildtype but not catalytic dead SIRT6 to the DUE of *Utrn* gene reduced *Utrn* transcription and erased H3K56ac locally. The results are now shown in the (revised Fig.6 c,d).

8. Would CK levels increase if the experiments in Fig. 6E and 6H were performed in the absence of the *mdx* mutation?

Response: WT myofibers do not release CK when dystrophin is functional and the sarcolemma is intact, which is also reflected by the very low serum CK levels of control mice as shown in (Fig. 3f). We measured CK activity in culture medium of myotubes derived from *mdx* and *Sirt6*^{mKO}/*mdx* MuSCs as a proxy for partially restored membrane integrity. Inactivation of Sirt6 causes no obvious defects in myofiber membrane integrity. Forced deacetylation of H3K56ac by recruitment of SIRT6 to DUE of *Utrn* gene results in decreased expression of *Utrn*, which will have no effects on CK levels in the medium when dystrophin is functional and the membrane is intact. Changes in CK levels can only be expected when expression of *Utrn* matters, i.e. in *mdx* muscle cells. Thus, it does not make sense to measure CK levels in muscle cells, in which dystrophin is functional and the membrane is intact. We hope the reviewer shares this view.

9. The experiment in Figure 7 is also very interesting. However, these mice are very poorly characterised. What is their body size, fat mass, muscle mass, muscle force, etc.? It might simply be that the mice are in a very poor shape when they lack both the *mdx* and the *Utrn* genes. There were multiple gene expression changes that were affected by the SIRT6 mKO irrespectively of the *mdx* mutation. Does the *Utrn* mutation affects them?

Response: We complied to the reviewer's request and analyzed body size, muscle mass, and fat mass of control, *Utrn*^{-/-}/*mdx* and *Sirt6*^{mKO}/*Utrn*^{-/-}/*mdx* mice. *Utrn*^{-/-}/*mdx* and *Sirt6*^{mKO}/*Utrn*^{-/-}/*mdx* mice are smaller compare to controls. MRI measurements revealed a significant reduction of fat but not of muscle volume in both *Utrn*^{-/-}/*mdx* and *Sirt6*^{mKO}/*Utrn*^{-/-}/*mdx* mice. The results are consistent with published data showing that *Utrn*^{-/-}/*mdx* mice exhibit progressive weight loss, and muscle weakness by 4–6 weeks of

age¹⁵. Surprisingly, body weight to tibia length ratios are not significantly different between control and *Utrn*^{-/-}*mdx* and *Sirt6*^{mKO}/*Utrn*^{-/-}*mdx* mice and the TA muscle weight to tibia length ratio was even higher in *Utrn*^{-/-}*mdx* and *Sirt6*^{mKO}/*Utrn*^{-/-}*mdx* mice compared to controls, suggesting that the reduction in body growth is more severe than the reduction in muscle weight. These results are now shown in the (revised Supplemental Fig. 8a-c). In our hands, *Utrn*^{-/-}*mdx* mice die prematurely most often before 10 weeks of age due to respiratory or cardiac failure¹⁵. We assume that retardation in body growth is mainly due to respiratory and heart problems. Unfortunately, we are not able to assess the contribution of reduced cardiac function, since no mouse strain is available that specifically lacks *Utrn* expression in skeletal muscles but not the heart. We did not measure muscle force, since *Utrn*^{-/-}*mdx* mice are very hard to breed and usually die before they generate progeny. Only a small cohort of mice was available, which was not suitably sized for reliable force measurement.

We thank the reviewer for the opportunity to clarify the gene expression changes in the different mouse lines. Actually, we did not describe the number of deregulated genes in *Sirt6*^{mKO} compared to control mice in the previous version of the manuscript. We have now included these data in (revised Supplemental Fig. 2). In total, 189 genes were upregulated in *Sirt6*^{mKO} compared to control mice, of which only 9 genes changed expression when dystrophin was mutated (revised Supplemental Fig. 3g). We only generated *Sirt6*^{mKO}/*mdx* and *Sirt6*^{mKO}/*Utrn*^{-/-}*mdx* mice but not *Sirt6*^{mKO}/*Utrn*^{-/-} mice and therefore cannot directly answer the question whether mutation of *Utrn* has an impact on the 189 genes that were upregulated in *Sirt6*^{mKO} mice. However, we reason that mutation of *Utrn* has no major effect on the 189 genes that are upregulated in *Sirt6*^{mKO} mice compared to controls, since even mutation of dystrophin affects only 9 genes. Of course, the situation is very different when *Sirt6*^{mKO}/*mdx* and *mdx* mice are compared. Due to the increased stability of *mdx* myofibers numerous secondary events caused by myofiber instability do not happen in MuSCs anymore, which is the reason why the number of 682 upregulated genes declines substantially in *Sirt6*^{mKO}/*mdx* MuSC (revised Supplemental Fig. 3f).

Minor:

- *SIRT7* mRNA levels in MuSCs from *mdx* mice are highly unregulated. Is this consistent at the protein level?

Response: The bar graphs in the initial submission suggested increased expression of *Sirt7* mRNA in *mdx* muscle cells. However, the differences were mainly caused by an outlier, which also rendered the differences in gene expression non-significant. We have redone the RT-qPCR expression analysis using newly prepared samples (n=4). RT-qPCR result revealed that *Sirt7* mRNA levels in *mdx* mice are slightly increased without statistical significance. (Fig.1c) was revised accordingly. We also analyzed SIRT7 protein levels in *mdx* compared to control mice (Figure 4 to the viewer) and did not observe any significant differences. Since *Sirt7* is not in the focus of this study, we do not want to include this result in the manuscript.

Figure 4 for reviewer: Western blot analysis of SIRT7 level in MuSCs from control and *mdx* mice. GAPDH was used as loading control.

- In the muscle regeneration experiment shown in Supplemental Figure 1, a more detailed time course experiment would be needed, as the 14 day timing might not reflect a slower (even if effective) regeneration.

Response: We agree that it is important to rule out that inactivation of *Sirt6* has an impact on muscle regeneration. Thus, we subjected *Sirt6*^{mKO} mice to 3 rounds of cardiotoxin (CTX)-induced skeletal muscle regeneration with 21 days of recovery time between individual CTX injections and then assessed

skeletal muscle regeneration 7 days after the last CTX injection. Based on H&E staining and muscle weight measurements, we did not find any hints for impaired muscle regeneration in *Sirt6*^{mKO} compared to WT mice (revised supplemental Fig. 2k). In contrast, TA muscle mass was even slightly higher in *Sirt6*^{mKO} compared to WT mice, suggesting that Sirt6 inactivation accelerates rather than slows down muscle regeneration after injury (revised supplemental Fig. 2j).

- Line 144. Where do the 1285 genes come from? The graph depicts 977.

Response: The reviewer is right. The correct number of upregulated genes is 977. We have corrected this mistake. Thank for the careful reading.

- Lines 255-256. See point 6. The fact that H3K56 acetylation is upregulated at the DUE of the *Utrn* gene in *SIRT6* mKO/mdx mice does not proof that *SIRT6* is the enzyme directly performing this deacetylation reaction.

Response: To confirm that SIRT6 is the enzyme directly deacetylating H3K56ac within the DUE of *Utrn*, we performed ChIP-qPCR experiments. We observed a high enrichment of SIRT6 at the DUE of the *Utrn* gene using chromatin from C2C12 muscle cells, which is now shown in the (revised Fig. 5k). In addition, we also did ChIP-qPCR using chromatin from primary muscles cells expressing HA-SIRT6, since the quality of the antibodies is not sufficient to detect endogenous SIRT6 in ChIP experiments. Both experimental approaches demonstrate binding of SIRT6 to the DUE of the *Utrn* gene, which in combination with the increase of H3K56ac after inactivation of *Sirt6* makes it very likely that SIRT6 directly deacetylates H3K56ac in this region (Figure 5k, l).

- What is the expression of the *Utrn* gene in *SIRT6* mKO mice compared to control, in the absence of the mdx mutation?

Response: We have determined protein levels of utrophin protein in *Sirt6*^{mKO} muscle tissue by western blot analysis. We found a significantly increase of UTRN after *Sirt6* inactivation. The results are shown in the revised (Supplemental Fig. 6a).

References

1. Wust, S. *et al.* Metabolic Maturation during Muscle Stem Cell Differentiation Is Achieved by miR-1/133a-Mediated Inhibition of the Dlk1-Dio3 Mega Gene Cluster. *Cell Metabolism* **27**, 1026-+ (2018).
2. Haring, M. *et al.* Chromatin immunoprecipitation: optimization, quantitative analysis and data normalization. *Plant Methods* **3**(2007).
3. Bernier, M. *et al.* Linker histone H1 and H3K56 acetylation are antagonistic regulators of nucleosome dynamics. *Nature Communications* **6**(2015).
4. Pratt, S.J.P., Valencia, A.P., Le, G.K., Shah, S.B. & Lovering, R.M. Pre- and postsynaptic changes in the neuromuscular junction in dystrophic mice. *Frontiers in Physiology* **6**(2015).
5. Baumann, C.W., Warren, G.L. & Lowe, D.A. Plasmalemma Function Is Rapidly Restored in Mdx Muscle after Eccentric Contractions. *Medicine and Science in Sports and Exercise* **52**, 354-361 (2020).
6. Zhong, L. *et al.* The histone deacetylase Sirt6 regulates glucose homeostasis via Hif1alpha. *Cell* **140**, 280-93 (2010).
7. Ryall, J.G. *et al.* The NAD(+)-Dependent SIRT1 Deacetylase Translates a Metabolic Switch into Regulatory Epigenetics in Skeletal Muscle Stem Cells. *Cell Stem Cell* **16**, 171-183 (2015).
8. Sreenivasan, K. *et al.* Attenuated Epigenetic Suppression of Muscle Stem Cell Necroptosis Is Required for Efficient Regeneration of Dystrophic Muscles. *Cell Reports* **31**(2020).
9. Yamaguchi, M. *et al.* Calcitonin Receptor Signaling Inhibits Muscle Stem Cells from Escaping the Quiescent State and the Niche. *Cell Reports* **13**, 302-314 (2015).
10. Lee, K.S. *et al.* Runx2 is a common target of transforming growth factor beta 1 and bone morphogenetic protein 2, and cooperation between Runx2 and Smad5 induces osteoblast-specific gene expression in the pluripotent mesenchymal precursor cell line C2C12. *Molecular and Cellular Biology* **20**, 8783-8792 (2000).
11. Asakura, A., Komaki, M. & Rudnicki, M.A. Muscle satellite cells are multipotential stem cells that exhibit myogenic, osteogenic, and adipogenic differentiation. *Differentiation* **68**, 245-253 (2001).
12. Biressi, S., Miyabara, E.H., Gopinath, S.D., Carlig, P.M. & Rando, T.A. A Wnt-TGFbeta2 axis induces a fibrogenic program in muscle stem cells from dystrophic mice. *Sci Transl Med* **6**, 267ra176 (2014).
13. Even, P.C., Decrouy, A. & Chinet, A. Defective regulation of energy metabolism in mdx-mouse skeletal muscles. *Biochem J* **304 (Pt 2)**, 649-54 (1994).
14. Moore, T.M. *et al.* Mitochondrial Dysfunction Is an Early Consequence of Partial or Complete Dystrophin Loss in mdx Mice. *Frontiers in Physiology* **11**(2020).
15. Grady, R.M. *et al.* Skeletal and cardiac myopathies in mice lacking utrophin and dystrophin: A model for Duchenne muscular dystrophy. *Cell* **90**, 729-738 (1997).

REVIEWERS' COMMENTS

Reviewer #2 (Remarks to the Author):

The authors have responded well to the comments with extra data and analysis. This has greatly strengthened the paper.

Reviewer #3 (Remarks to the Author):

The authors have done an outstanding job in tackling the main and minor points raised during my initial revision.

Responses to reviewers

Reviewer #1:

Please note that Reviewer #1 did not leave a formal report, but in their confidential comments indicated that the concerns originally raised were sufficiently addressed.

Response: We appreciate the important comments by the reviewer and are glad that we solved all open issues.

Reviewer #2 (Remarks to the Author):

The authors have responded well to the comments with extra data and analysis. This has greatly strengthened the paper.

Response: We are delighted about the positive reception of the revised paper.

Reviewer #3 (Remarks to the Author):

The authors have done an outstanding job in tackling the main and minor points raised during my initial revision.

Response: We appreciate that the reviewer acknowledges that we “...have done an outstanding job in tackling the main and minor points raised during my initial revision”.